# IQ-Switch is a QF-based innocuous, silencing-free, and inducible gene switch system in zebrafish

Jeongkwan Hong[1,11], Jae-Geun Lee[2,3,4,11], Kyung-Cheol Sohn[5,11], Kayoung Lee[1], Seoee Lee[1], Jinyoung Lee[1], Jihye Hong[1], Dongju Choi[1], Yeseul Hong[1], Hyo Sun Jin[6], Dae-Kyoung Choi[6], Su Ui Lee[7], Yun Kee[8], Jangham Jung[1], Young-Ki Bae[9], Ran Hee Hwang[10], Gang Min Hur [5✉], Jeong-Soo Lee [2,3,4✉] & Hyunju Ro [1✉]

Though various transgene expression switches have been adopted in a wide variety of organisms for basic and biomedical research, intrinsic obstacles of those existing systems, including toxicity and silencing, have been limiting their use in vertebrate transgenesis. Here we demonstrate a novel QF-based binary transgene switch (IQ-Switch) that is relatively free of driver toxicity and transgene silencing, and exhibits potent and highly tunable transgene activation by the chemical inducer tebufenozide, a non-toxic lipophilic molecule to developing zebrafish with negligible background. The interchangeable IQ-Switch makes it possible to elicit ubiquitous and tissue specific transgene expression in a spatiotemporal manner. We generated a RASopathy disease model using IQ-Switch and demonstrated that the RASopathy symptoms were ameliorated by the specific BRAF$^{(V600E)}$ inhibitor vemurafenib, validating the therapeutic use of the gene switch. The orthogonal IQ-Switch provides a state-of-the-art platform for flexible regulation of transgene expression in zebrafish, potentially applicable in cell-based systems and other model organisms.

[1] Department of Biological Science, College of Biosciences and Biotechnology, Chungnam National University, Daejeon, Republic of Korea. [2] Disease Target Structure Research Center, KRIBB, 125 Gwahak-ro, Yuseong-gu, Daejeon 34141, Republic of Korea. [3] Department of Functional Genomics, KRIBB School, University of Science and Technology, 217 Gajeong-ro, Yuseong-gu, Daejeon 34113, Republic of Korea. [4] Dementia DTC R&D Convergence Program, KIST, Hwarang-ro 14 gil 5, Seongbuk-gu, Seoul 02792, Republic of Korea. [5] Department of Pharmaclogy, College of Medicine, Chungnam National University, Daejeon, Republic of Korea. [6] Biomedical Research Institute, Chungnam National University Hospital, 282, Munhwa-ro, Jung-gu, Daejeon 35015, Republic of Korea. [7] Natural Medicine Research Center, Korea Research Institute of Bioscience and Biotechnology (KRIBB), Cheongju, Republic of Korea. [8] Division of Biomedical Convergence, College of Biomedical Science, Kangwon National University, Chuncheon, Republic of Korea. [9] Division of Cancer Biology, Research Institute, National Cancer Center, Goyang-si, Republic of Korea. [10] Department of Nursing, Gwangju Women's University, Gwangju, Republic of Korea. [11]These authors contributed equally: Jeongkwan Hong, Jae-Geun Lee, Kyung-Cheol Sohn. ✉email: gmhur@cnu.ac.kr; jeongsoo@kribb.re.k; rohyunju@cnu.ac.kr

Though a variety of inducible transgene expression switches (ITESs) with binary features have made a substantial impact on scientific progress for understanding basic cellular functions of genes of interest, discrete limitations of hitherto developed ITESs have been preventing their wider applications at the whole-organism level, in particular in vertebrates[1,2]. A common ITES is composed of two elements. One is a driver construct, which encodes a promoter for the enforced expression of a stimulus-responsive heterologous transcriptional activator (TA), and the other is an effector construct that consists of tandem repeats of TA binding elements followed by basal transcription-initiating sequences required for the transgene expression through the input of an adequate stimulus. The stimulus can be a specific chemical or pleiotropic stress, including heat shock, heavy metals, or hormones.

Among the various binary ITESs, tetracycline-dependent transactivator (Tet-On/Off) and GAL4/UAS-based transgene switches have been most frequently used for the generation of genetic model organisms[2]. Especially, a sophisticatedly designed heterologous GAL4-TA composed of an isolated GAL4 DNA-binding domain and a minimal VP16 activation domain fused to a modified ecdysone-binding region of EcR had been adopted in zebrafish for the controlled expression of transgenes which could be tightly regulated by several different ecdysone agonists[3]. However, the relatively high leakiness of Tet-On/Off[4] and the silencing of GAL4-responsive UAS element due to accumulated methylations on its CpG dinucleotides in vertebrates[5] have been potentially problematic the use of these systems in transgenesis. The leakiness of ITESs could be completely plugged by concomitantly exploiting inducible Flp/FRT or Cre/LoxP DNA recombinase-based systems to initiate transgene expression by the excision of regulatory elements[6,7]. However, as a DNA recombinase-based ITES becomes completely irreversible once stimulated, it may not be ideal when tunable or stepwise transgene expression is preferable. To overcome the methylation barrier in the GAL4/UAS-based system, a CpG-free element (tUAS) was adopted in zebrafish with a novel orthogonal driver (TrpR) but nonetheless, the unbearable toxicity of the driver limited its application in transgenesis[8].

Other currently available binary ITESs are valuable for the generation of transgenic animals, however, each has also inherent drawbacks. For instance, the chemical stimulators mifepristone and rapamycin, which have been used in the LexPR-LexOP[9,10], and dimerizer[1,11] systems, respectively, have potential adverse effects on reproduction and embryonic development. Moreover, as the dimerizer system becomes irreversible after treatment with rapamycin, it would not be an optimal option if adjustable expression of transgene were necessary.

To extricate the transgene expression switch from gene silencing, the QF/QUAS binary expression system has been recently introduced to zebrafish transgenesis[12-14]. The Q system, which originated from a quinic acid-sensing system of Neurospora crassa, is composed of a TA (QF), five tandem repeats of QF-responsive element (5xQUAS), and a QF repressor (QS)[15,16]. The QUAS element can be derepressed by the addition of quinic acid, which interferes with the inhibitory effect of QS on QF[16]. Though the Q system has been shown to function in diverse organisms including Escherichia coli[17], Drosophila[16,18], Caenorhabditis elegans[19,20], and plants[21], several intrinsic limitations of the system have hampered its direct application in vertebrates. These limitations include high toxicity of the QF driver[12-14], non-functionality of quinic acid on QS in mammalian cells, which prevents derepression by quinic acid[16], and detrimental effects of quinic acid on zebrafish embryonic development[12]. However, despite the aforementioned drawbacks, the potentially non-silencing feature of QF/QUAS would still be advantageous over

the GAL4/UAS system[12]. Current efforts have been devoted to combining the Q system with other gene switches, such as Tet-On[20] and the bacterial Lac repressor system[22], to circumvent the problematic traits of the original Q system. To expedite the progress of transgenesis in zebrafish, we newly developed a more intuitive and reliable QF/QUAS-based orthogonal ITES, which is free of gene silencing at least up to seventh generations, equipped with a non-toxic driver, non-leaky, highly tunable for expression of the transgene by an innocuous chemical inducer, reversible, and able to trigger several-fold higher transgene expression than the GAL4/UAS system. Hence, the QF-based binary gene switch we report in this study overcomes pivotal problems of currently available ITESs, thus providing a valuable experimental platform for the regulated expression of transgenes in zebrafish, with the potential for applications in cell-based systems and other whole organisms.

## Results

**Refining a novel QF driver that is sensitive to exogenous chemical stimuli.** To apply Q-system-based ITESs in zebrafish transgenesis, we created various chimeric transactivators that are sensitive to chemical stimuli to drive luciferase expression under the control of 5xQUAS. Initially we fused a hormone-binding domain of insect ecdysone receptor (EcR) to the full length QF TA at its C-terminus to gain nuclear access only when the chimera binds to the ecdysone agonist, tebufenozide (Teb, also known as RH-5992), a lipophilic molecule that is non-toxic to developing zebrafish (Fig. 1a, b)[3,23,24]. The chimeric construct driven by the zebrafish ubiquitin promoter[25] was highly effective in inducing luciferase expression in HEK293 cells; nonetheless, it was too toxic to adopt for zebrafish transgenesis (Fig. 1a-c, Supplementary Fig. 1 and Supplementary Table 1). Though the middle domain truncated form of QF was reported to be non-toxic in Drosophila[26], it was still detrimental to zebrafish embryos and unacceptably leaky (Fig. 1a-c, Supplementary Fig. 1 and Supplementary Table 1). Therefore, we modified the original QF until we obtained a construct that could suppress the intrinsic toxicity while retaining the functional aspect as a TA. The QF is composed of a DNA-binding domain (QFDBD) at its N-terminus, followed by a minimal activation domain (AD*) that is regarded as an N-terminal part of a middle domain (MD), and a main transcriptional activation domain (AD) (Fig. 1a)[26]. Among the several driver constructs we tested, there was only one orthogonal configuration selected for further assays. The selected driver was composed of a QFDBD, two tandem repeats of AD*, and a minimal VP16 transactivator (VP16*)[3] followed by EcR (QFDBD-2xAD*-VP16*-EcR, Driver 3 in Fig. 1a-c). In zebrafish embryos the driver did not elicit significant toxicity, irrespective of Teb treatments (Supplementary Figs. 1, 2 and Supplementary Table 1). In addition, though relatively weaker than the original QF, the driver (hereafter IQ-Switch, Inducible QF transgene expression Switch) responded well to Teb without gross luciferase leakiness (Fig. 1a-c and Supplementary Table 1). An alternative was a QF-Gal4 heterologous driver (QFDBD-Gal4AD), which was recently reported to show a tolerable toxicity but capable of stimulating transgene located downstream of 5xQUAS in zebrafish[14]. However, in comparison with QFDBD-2xAD*-VP16*, QF-Gal4 was relatively more toxic in proportion to the amount of transcripts whose expression elicited eccentric embryonic defects, including headless and severe ventralization, which were not observed with comparable injection of QFDBD-2xAD*-VP16* mRNA (Supplementary Table 1 and Supplementary Fig. 2).

The IQ-Switch translocated into the nucleus after administration of Teb, and then turned on the mCherry reporter gene

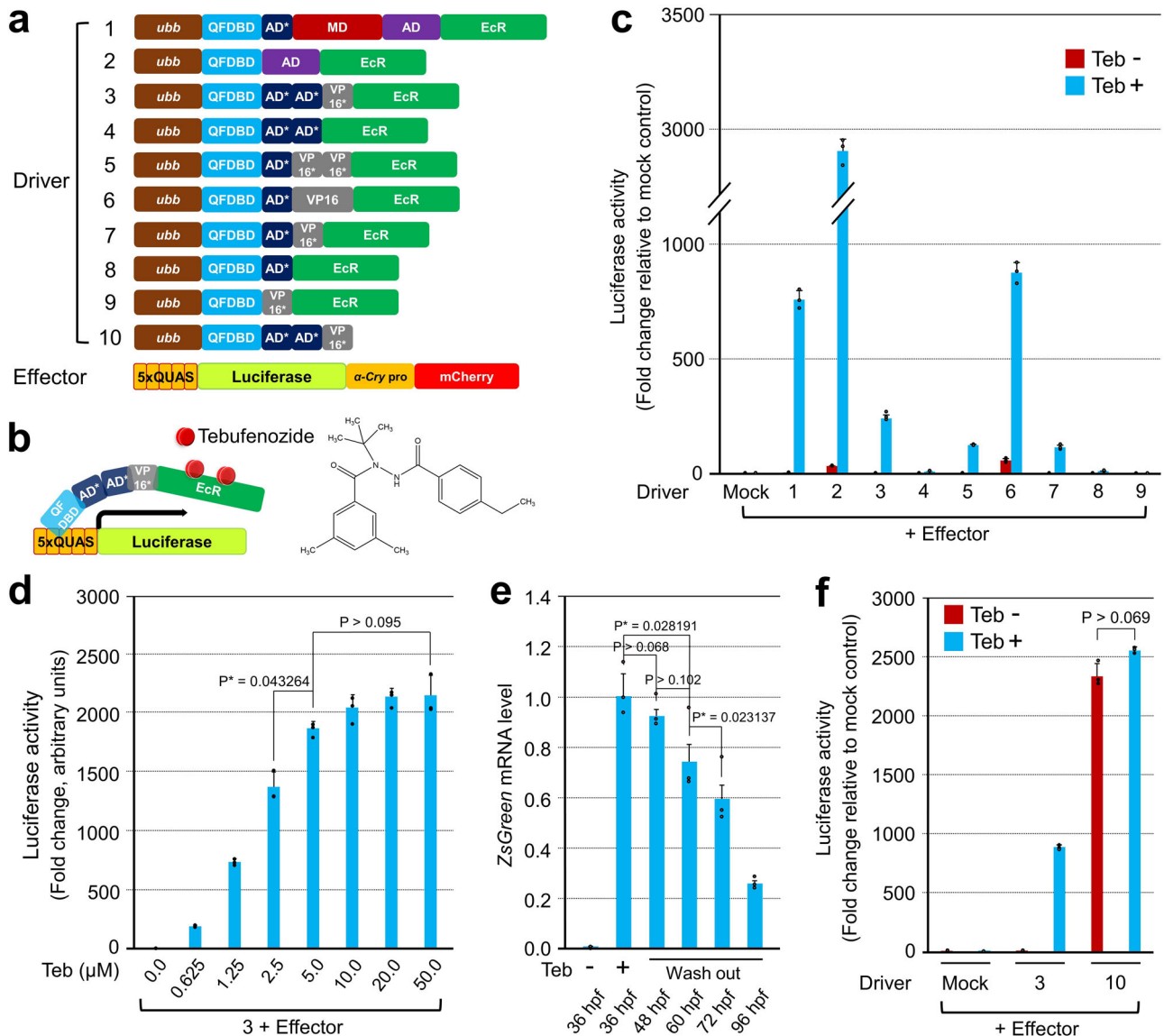

**Fig. 1 Optimization of IQ-Switch for toxicity and leakiness, while retaining transgene-stimulating activity. a** Alignment of driver and effector constructs tested for the optimization of IQ-Switch. **b** Schematic drawing of an executing mechanism of IQ-Switch. The chemical structure of tebufenozide is depicted on the right. **c** Measurement of luciferase reporter activity after transfection of an individual driver plasmid, indicated by number, together with the same effector containing 5xQUAS elements. After 24 h, the transfected HEK293 cells were exposed to 10 μM of Teb for another 24 h before measurement. The luciferase reporter activity responding to Teb is indicated in azure; the basal leakiness of the effector by a specific inducer plasmid is visualized in reddish brown. **d** The combination of driver '3' in (**a**) with the effector in HEK293 cells showed responsiveness in a Teb dosage-dependent manner. Error bars stand for the standard deviation. **e** Reversibility of IQ-Switch. The offspring of the genetic cross between $Tg(ubb:QFDBD-2xAD^*-VP16^*-EcR)$ and $Tg(5xQUAS:ZsGreen-P2A-lamin A^{\Delta 37})$ were subjected to RT-qPCR. After 50 μM of Teb was used to treat embryos at 12 hpf for 24 h, selected ZsGreen-positive embryos were cultured under the Teb-minus condition for the indicated period. Error bars stand for the standard error of the mean. **f** Measurement of luciferase activity in HEK293 cells after transiently transfecting the indicated combination of plasmids. The experimental conditions are identical to those in (**c**). Driver '10' in (**a**) without EcR strongly stimulated the luciferase reporter activity irrespective of the presence of Teb, the difference of which was statistically non-significant. Luciferase activity was measured in triplicate or more. *P*-values were indicated in each figure.

located after 5xQUAS in HEK293 cells (Supplementary Fig. 3a–j). The activity of the luciferase reporter was gradually potentiated by the increased dosage of Teb until it reached a plateau at 5 μM of Teb in cell culture medium (Fig. 1d). The Teb dosage-dependent responsiveness of IQ-Switch was corroborated again in zebrafish embryos by western blotting. The IQ-Switch F2 driver, $Tg(ubb:QFDBD-2xAD^*-VP16^*-EcR)$, was crossed with the F2 effector, $Tg(13xQUAS:ZsGreen-P2A)$, to obtain F3 siblings that were then exposed to different concentrations of Teb. Similar to the cell culture results (Fig. 1d), we observed that ZsGreen expression levels gradually increased

in proportion to the concentration of Teb (Supplementary Fig. 4). To test whether the IQ-Switch was reversible by the withdrawal of the chemical inducer, we generated a transgenic zebrafish carrying a mutated *lamin A (Δ37)* gene causative of Hutchinson-Gilford progeria syndrome[27,28]. As expected, a genetic cross between the $Tg(ubb:QFDBD-2xAD^*-VP16^*-EcR)$ driver line (F2) and $Tg(5xQUAS:ZsGreen-P2A-lamin A^{\Delta 37})$ effector line (F3) produced embryos reactive to Teb (50 μM) with expected Mendelian inheritance ratios, with ~25% of embryos showing ZsGreen expression. The ZsGreen-positive embryos were collected at the indicated time points in Fig. 1e for

the preparation of cDNA, which was then subjected to quantitative polymerase chain reaction (RT-qPCR) with primer sets specific for zebrafish $\beta$-actin and ZsGreen. The yield of transcripts encoding ZsGreen-P2A-lamin $A^{\Delta 37}$ was gradually reduced after Teb removal, indicating the reversibility of IQ-Switch (Fig. 1e). The relatively slow off-kinetics of Teb corresponded well with the previous reports[4,23,29] showing that it took time to completely metabolize ecdysteroids and their agonists in cells. In mice, the ecdysteroids are taken up and metabolized by the liver before excretion through both urinary and fecal routes[23]. Another merit of IQ-Switch is that by removing EcR, the tunable transactivator becomes diverted into a constitutively active form (hereafter EQ-On, Everlasting QF transgene switch-On), where its configuration no longer requires Teb stimulation for constant transgene induction (Driver 10, in Fig. 1a). The EQ-On was mainly localized in the nucleus (Supplementary Fig. 3b, b′, k-r, k′-r′) and more than two-fold stronger than IQ-Switch (Fig. 1f) with Teb stimulation, with minimal toxicity (Supplementary Figs. 1 and 2). The potential use of the switch was validated in zebrafish, in which the progeny obtained from the genetic cross of Tg(ubb:QFDBD-2xAD*-VP16*) and Tg(5xQUAS:ZsGreen-P2A-lamin $A^{\Delta 37}$) showed ZsGreen reporter expression as early as 6 h post fertilization (hpf) without Teb administration (Supplementary Fig. 5).

**Methylation-independent activation of IQ-Switch.** A previous report showed that the existence of multiple CpG dinucleotides in 5xQUAS did not interfere with GFP reporter expression by the stimulation of a QF-Gal4 heterologous driver[14], and that artificial CpG-free 5xQUAS could not elicit further strong fluorescent reporter gene expression over the natural 5xQUAS[14]. Thus, we hypothesized that the natural feature of CpG dinucleotides in 5xQUAS would be refractory to methylation. To identify the methylated state of the 5xQUAS element, we carried out bisulfite sequencing analysis of the F4 generation of effector zebrafish having a human BRAF(V600E) transgene under the control of 5xQUAS, Tg(5xQUAS:ZsGreen-P2A-BRAF(V600E)). As a control, we used the F2 effector line with the zebrafish braf(V610E) transgene under the control of 10xUAS element, Tg(10xUAS:ZsGreen-P2A-braf(V610E)), and Tg(ubb:Gal4-VP16*-EcR), which is responsive to a Gal4 driver[3] with Teb stimulus. Interestingly, methyl groups were heavily deposited on CpG dinucleotides in both elements (Fig. 2a), suggesting that activation of the 5xQUAS element occurs irrespective of its methylation. To evaluate whether the IQ-Switch driver could bind to methylated QUAS to trigger transgene expression, we stimulated embryos obtained from genetic crosses between driver and effector lines with 50 μM of Teb for only 2 h to elude early embryonic lethality of braf(V610E) and BRAF(V600E) transgenes[30] (Fig. 2b, Supplementary Fig. 6a). Indeed, enforced expression of drivers by injection of mRNA encoding QFDBD-2xAD*-VP16*-EcR with a 6xMyc epitope at its N-terminus into the Tg(5xQUAS:ZsGreen-P2A-BRAF(V600E)) effector (F5 generation) caused distinctive embryonic malformation when Teb (50 μM) was administered for the indicated time window (Fig. 2b, c). At the molecular level, through the chromatin immunoprecipitation (ChIP) assay, we found that enrichment of the 6xMyc-tagged driver on the 5xQUAS elements, but uninjected or 6xMyc-Gal4DBD-VP16*-EcR mRNA-injected control embryos did not elicit statistically significant levels of chromatin precipitation under identical experimental conditions (Fig. 2b, c). The enrichment of IQ-Switch driver on the methylated 5xQUAS was not a unique feature of QFDBD-2xAD*-VP16*-EcR because we also observed a similar enrichment of 6xMyc-QFDBD-Gal4AD-EcR[14] on the 5xQUAS under identical

experimental settings with mRNA encoding 6xMyc-QFDBD-Gal4AD-EcR instead of 6xMyc-QFDBD-2xAD*-VP16*-EcR (Supplementary Fig. 6). Therefore, the data strongly suggest that QFDBD binds to its specific cis-regulatory elements irrespective of its methylation status.

Moreover, while a genetic cross of F3 Tg(ubb:QFDBD-2xAD*-VP16*-EcR) driver and F6 Tg(5xQUAS:ZsGreen-P2A-BRAF(V600E)) effector with exposure of the embryos to Teb (50 μM) resulted in ubiquitous ZsGreen reporter expression and severe embryonic malformation in F7 progeny, mating F2 Tg(ubb:Gal4DBD-VP16*-EcR) driver with F2 Tg(10xUAS:ZsGreen-P2A-braf(V610E)) effector resulted in F3 embryos with variegated expression of reporter genes and relatively mild embryonic defects upon identical Teb treatment (Fig. 2e). Other representative embryos were included in Supplementary Fig. 7. Collectively these data strongly suggest that the QF DNA-binding domain stimulates transgene expression through direct binding to the QUAS element irrespective of its methylation state.

**Controlled expression of BRAF(V600E) manifested discrete RASopathy pathologies ameliorated by a selective BRAF inhibitor.** A germ line-transmitted animal that ubiquitously expresses BRAF(V600E) cannot be maintained due to the high toxicity of the BRAF(V600E) gain of function (GOF) mutation. Though a Cre/LoxP-based conditional knock-in mouse model with the BRAF(V600E) oncogene has been available for more comprehensive studies of disease progression[31], the CMV-Cre-mediated ubiquitous expression of BRAF(V600E) caused prenatal lethality, which hampered the design of strategies for therapeutic approaches. To test whether IQ-Switch can be used as a new drug-screening platform, controlled temporal expression of human BRAF(V600E) oncogene[32] was carried out by a genetic cross between Tg(ubb:QFDBD-2xAD*-VP16*-EcR) and Tg(5xQUAS:ZsGreen-P2A-BRAF(V600E)), followed by a low dose (2.5 μM) of Teb treatment at 36 hpf for 24 h, with or without administration of vemurafenib (15 μM), an FDA approved specific inhibitor of BRAF(V600E), for 36 h (Fig. 3a). As expected, triggering BRAF(V600E) expression at a later stage of zebrafish embryonic development was sufficient to induce visible malignancies such as craniofacial deformities, cardiac malformation, and cutaneous abnormalities, all of which were reminiscent of cardio-facio-cutaneous (CFC) syndrome, a RASopathy commonly caused by BRAF mutations[33–36] (Fig. 3i–p and Supplementary Fig. 8a–j). However, early induction of BRAF(V600E) before gastrulation caused severe embryonic malignancy with truncated posterior structure and compromised forebrain without noticeable eye structures at 24 hpf (Supplementary Fig. 8b, k, l). Strikingly, the CFC syndrome-like phenotypes by ectopic expression of BRAF(V600E) were profoundly ameliorated by simultaneous treatment with vemurafenib (Fig. 3q–w). Our data strongly suggest that the GOF disease models generated by our IQ-Switch can be used as a reliable and amenable drug-screening platform to help design more sophisticated therapeutic approaches.

**Optimization of QUAS tandem repeats to augment transgene activity.** Though we demonstrated that IQ-Switch with 5xQUAS has significant advantages over other ITESs, its responsiveness to Teb was relatively weak compared to the Gal4DBD-VP16*-EcR/10xUAS system (Fig. 4a, b)[3]. To improve sensitivity to Teb and to increase the transgene expression level of IQ-Switch, we initially tried to modify the driver construct by replacing its transcriptional activating module with other more potent activators (Supplementary Table 1). However, all of our efforts were futile because all transcriptional activating modules tested other than QFDBD-2xAD*-VP16*-EcR showed at least one fatal flaw rendering them inadequate for their application to transgenesis

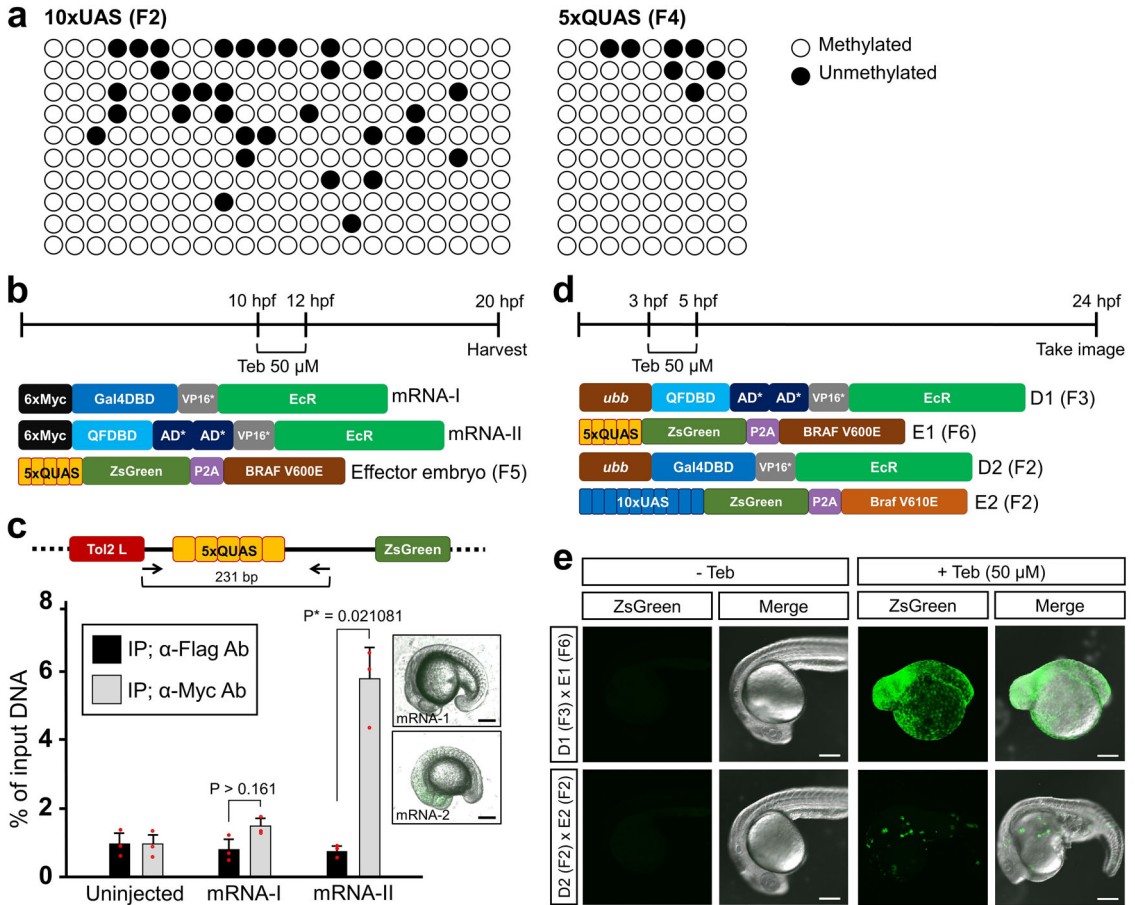

**Fig. 2 The activity of IQ-Switch is not influenced by QUAS methylation. a** The prepared genomic DNA from F2 offspring of *Tg(10xUAS:ZsGreen-P2A-braf^V610E^)* and F4 offspring of *Tg(5xQUAS:ZsGreen-P2A-BRAF^V600E^)* was subjected to bisulfite sequencing analysis. Both element regions became heavily methylated through successive generations. Open circles indicated methylated CpGs and dark circles represented unmethylated CpGs. Ten individual clones from each group were independently sequenced as shown on the vertical axis. **b** Schematic diagram of ChIP experiments in (**c**). The in vitro transcribed mRNA (50 pg) encoding *6xMyc*-tagged *Gal4DBD-VP16*-EcR* and *6xMyc-QFDBD-2xAD*-VP16*-EcR* was introduced into the embryos of *Tg(5xQUAS:ZsGreen-P2A-BRAF^V600E^)*. The injected embryos were exposed to 50 μM of Teb for 2 h at the early somite stage and then harvested at 20 hpf. **c** An amplicon encompassing the 5xQUAS element region occupies 231 bp. ChIP assay of embryo with anti-Myc and anti-Flag antibody. The anti-Flag antibody was used as a negative control. The ChIP samples were subjected to qPCR with the indicated primers (arrows). While mRNA encoding *6xMyc-Gal4DBD-VP16*-EcR* did not cause any embryonic malformation when exposed to Teb, sporadic exposure of *6xMyc-QFDBD-2xAD*-VP16*-EcR* mRNA-injected embryos to Teb showed developmental defects (boxes). q-PCR was carried out in triplicate, and standard errors of the mean are shown in the panel. Error bars stand for the standard deviation. **d** The embryos were treated with 50 μM of Teb for 2 h before the onset of gastrulation and then observed at 24 hpf. **e** While ZsGreen reporter under the control of QF/5xQUAS showed ubiquitous expression in the whole embryo (D1 + E1), the same reporter was activated in random tissues when driven by Gal4/10xUAS (D2 + E2). Note that 10xUAS regulated by Gal4 was progressively silenced as early as F3 generation. Abbreviations: Teb; tebufenozide. Scale bar; 200 μm.

(Supplementary Table 1). Thus, instead of altering the driver, we increased the length of the QUAS tandem repeats until a satisfying level of transgene expression was achieved by generating 5x, 9x, 13x, and 17xQUAS constructs (Fig. 4a and Supplementary Fig. 9). The luciferase reporter activity in HEK293 cells was gradually improved by increasing the number of QUAS elements with Teb stimulation. The 13xQUAS together with the IQ-Switch driver gave rise to strong *mCherry* reporter gene expression in Cos7 cells when treated with Teb (Supplementary Fig. 3a′–j′), and its activity increased by 7.3-fold over the original 5xQUAS (Fig. 4a, b). However, further increase of QUAS elements of up to 17 repeats failed to achieve stronger reporter activity, but rather compromised the luciferase activity on par with 9xQUAS (Fig. 4a, b). Strikingly, the combination of QFDBD-2xAD*-VP16*-EcR with 13xQUAS was ~4.5 fold more potent than Gal4DBD-VP16*-EcR/10xUAS in the luciferase assay (Fig. 4a, b) with negligible leakiness (Supplementary Fig. 10). The individual

IQ-Switch with indicated combinations of QUAS repeats, with the exception of 17xQUAS, was adapted to zebrafish transgenesis (Supplementary Fig. 11). The transgenic animals with ZsGreen reporter under the control of different tandem repeats of QUAS showed elevated expressivity of ZsGreen according to increased numbers of QUAS elements when crossed with *Tg(ubb:QFDBD-2xAD*-VP16*-EcR)* with a relatively low dosage of Teb (10 μM of Teb; Supplementary Fig. 11). To quantify transgene responsiveness to Teb (10 μM) stimuli, we measured the fluorescence strength of the embryos obtained from genetic crosses between an F3 *Tg(ubb:QFDBD-2xAD*-VP16*-EcR)* and discrete F2 effector lines with different numbers of QUAS with a *ZsGreen* reporter. To precisely measure the yield of fluorescence, we used a fixed confocal setting (see Materials for more detailed information) for the analysis of fluorescence intensity in the brain and muscle, and background as a negative control (Fig. 4c). The fluorescent signals gradually increased in proportion to the number of QUAS repeats

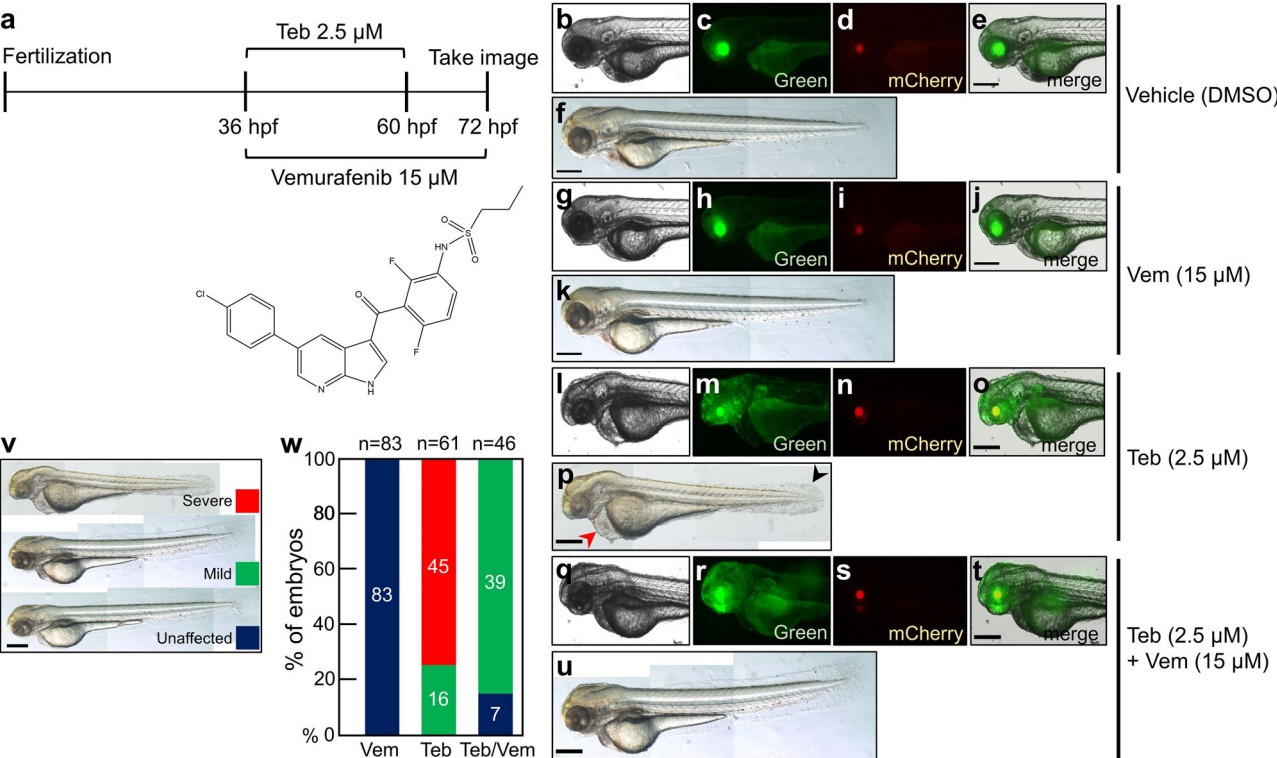

**Fig. 3 Modeling RASopathy disease with BRAF(V600E) overexpression using IQ-Switch. a** While a low dose of Teb (2.5 μM) was used to treat the genetic cross of *Tg(ubb:QFDBD-2xAD\*-VP16\*-EcR,α-cry:EGFP)* and *Tg(5xQUAS:ZsGreen-P2A-BRAFV600E,α-cry:mCherry)* from 36 hpf to 60 hpf, 15 μM of vemurafenib was added to the transgenic embryos for an extended period from 36 hpf to 72 hpf. The live embryos were observed under an epifluorescence microscope at 72 hpf. The chemical structure of vemurafenib is shown below. **b–u** While green fluorescence in the lens manifested the existence of a driver cassette, mCherry in the lens represented an effector *5xQUAS:ZsGreen-P2A-BRAFV600E* transgene in the genome. **b–f** DMSO vehicle treatment. **g–k** Treatment with 15 μM of vemurafenib alone. **l–p** Treatment with 2.5 μM of Teb alone. The embryos showed ZsGreen reporter expression in whole tissues. **q–u** A representative image of embryos exposed simultaneously to both Teb (2.5 μM) and vemurafenib (15 μM). Similar to that of Teb alone, green fluorescence was detected in all tissues. **v** Embryos were classified into three groups following embryonic malformation. **w** A graphical view of the embryonic defects as in (**v**). n represents the number of embryos analyzed. Abbreviations: Teb; tebufenozide, Vem; vemurafenib. Scale bar; 150 μm.

in both the Z-stacking (Fig. 4c, d) and single-plane images (Supplementary Fig. 12). However, with high levels of Teb stimulation (50 μM of Teb), the strength of ZsGreen fluorescence at least under the confocal microscope with 9xQUAS seemed to be reached a maximum level, comparable to that with 13xQUAS (Supplementary Fig. 13). Therefore, the expression level of IQ-Switch can be modulated at least by two different approaches in vivo: via the dosage of the chemical inducer, and via the number of QUAS tandem repeats.

The gradual increase in transgene expression according to the order of QUAS repeats was quantitatively revalidated by western blotting in zebrafish embryos obtained from genetic crosses between an F3 *Tg(ubb:QFDBD-2xAD\*-VP16\*-EcR)* driver line with F2 effector lines, *Tg(5xQUAS:ZsGreen-P2A)*, *Tg(9xQUAS:Zs-Green-P2A)*, and *Tg(13xQUAS:ZsGreen-P2A)*. The embryos from individual genetic crosses at 1 day post fertilization (dpf) were subsequently treated with 10 μM of Teb for 24 h and then subjected to western blot analysis using anti-ZsGreen antibody. Teb-stimulated embryos under the control of 13xQUAS showed by far the most elevated level of ZsGreen expression in comparison to that with fewer QUAS tandem repeats (Supplementary Fig. 14).

The different reactivities of the individual QUAS constructs were validated again by observing transient ZsGreen reporter expression in the transgenic zebrafish *Tg(ubb:QFDBD-2xAD\*-VP16\*)* through introduction of plasmids encoding discrete QUAS repeats upstream of *ZsGreen*. As expected, ZsGreen fluorescence was most intense under the control of 13xQUAS (Supplementary Fig. 15). Notably,

the combination of IQ-Switch with 13xQUAS showed an equivalent level of luciferase reporter activity to the other two potent driver configurations (QFDBD-2xAD\*-VP32-EcR and QFDBD-Gal4AD-EcR), whose activities in general reached the maximum intensity with 9xQUAS (Supplementary Fig. 16).

**Neuron-specific expression of ZsGreen with gradual intensities using IQ-Switch and EQ-On.** To test whether IQ-Switch was applicable for labeling specific cells or tissues, we substituted ~8.7 kb of *elavl3*[37] for the *ubb* promoter in the driver cassette to visualize pan-neuronal tissues (Fig. 5 and Supplementary Fig. 17). Importantly, genetic crosses between F1 *Tg(elavl3:QFDBD-2xAD\*-VP16\*-EcR)* and F2 *Tg* with *ZsGreen* transgene under the control of different numbers of QUAS (5x, 9x, 13x) activated ZsGreen expression with different strengths in the order of the length of QUAS repeats only when stimulated with 50 μM of Teb for 48 h from 3 dpf (Fig. 5a, b). When the offspring obtained from a similar genetic cross between a Teb-responsive driver and *Tg(5xQUAS:ZsGreen-P2A)* were exposed to the same dose of Teb for 24 h before imaging at 48 hpf, ZsGreen fluorescence was observed throughout the entire neuronal tissues, including axonal fibers innervating neuromasts, which completely recapitulates the well-known *elavl3* expression domains[38,39] (Supplementary Fig. 17). Even more striking was the EQ-On dependent expression of ZsGreen in neurons driven by the cross between *Tg(e-lavl3:QFDBD-2xAD\*-VP16\*)* and *Tg(QUAS:ZsGreen)* in which fluorescent signals were gradually rising in whole neurons,

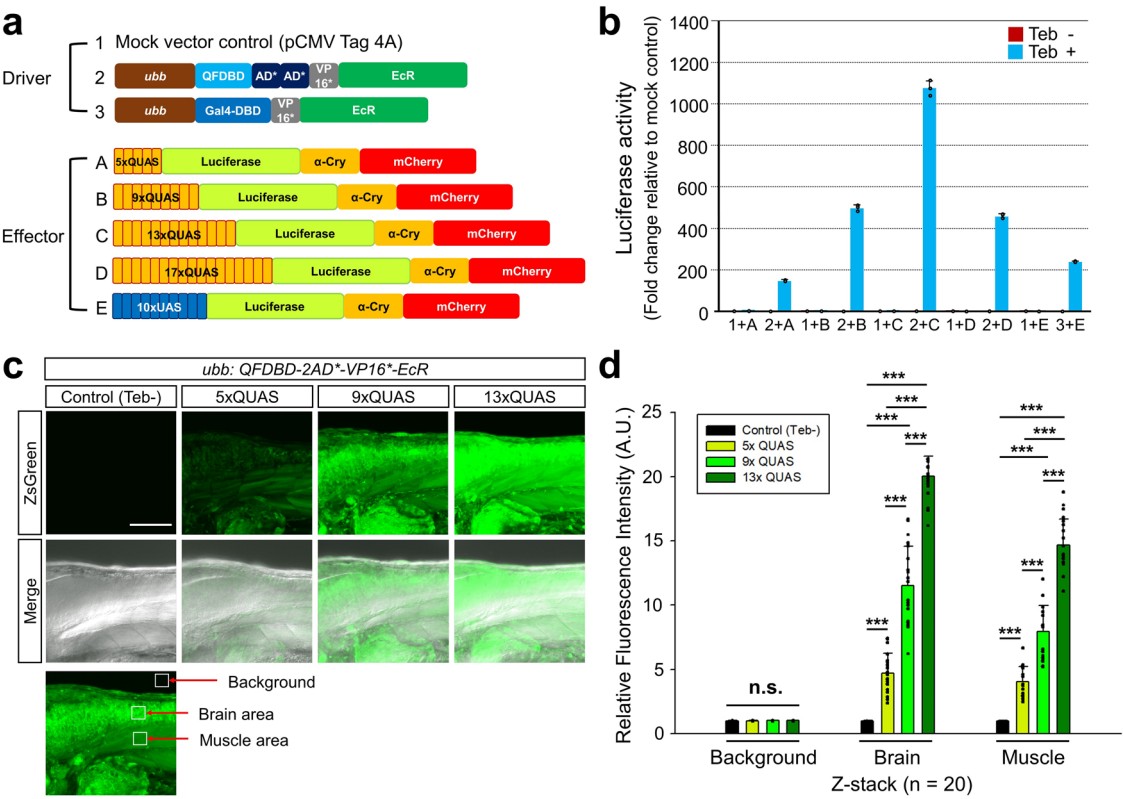

**Fig. 4 Optimization of QUAS repeats for stepwise transgene overexpression. a** Schematic drawing of driver and effector constructs. **b** Luciferase activities measured in HEK293 cells transiently transfected with the indicated driver and effector combinations. HEK293 cells were treated with 10 µM of Teb 24 h after transfection and then raised for another 24 h before harvest. Note that the combination of QF ("2") and 13xQUAS ("C") was the best at potentiating transgene induction. **c** Representative Z-stacking image of embryos at 3 dpf exposed to 10 µM of Teb for 24 h. Embryo harboring both the driver and 13xQUAS-regulated ZsGreen effector without treatment of Teb was used as a negative control, which could be easily selected by observation of red and green fluorescence in the lens. **d** The areas demarcated by open white boxes in (**c**) represented relative fluorescent intensities in the brain and muscle being evaluated in comparison to the level of control background intensity without Teb treatment, which was assumed as 1 arbitrary unit. Z-stacking images from randomly collected embryos (n = 20) were used to measure the expression level of ZsGreen regulated by 5x, 9x, and 13xQUAS. Error bars stand for the standard deviation. P-value indicated with (***) stands for < 0.001. Abbreviations: A.U.; arbitrary unit, n.s.; not significant, Teb; tebufenozide. Scale bar: 100 µm.

including sensory neurons in the caudal fin where the ZsGreen became most intensified by far with 13xQUAS (Fig. 5c, I-III). As the caudal fin axon branches are not usually observed in conventional Tg(elavl3:EGFP)[38], our modified QF/QUAS transgene induction systems could be used to unveil weak expression domains by potentiating the promoter activity.

The process of neuronal development of growing embryos can be analyzed by labeling neurons with appropriate fluorescent markers. Therefore, we set out to trace developing sensory neurons in epithelial cell layers in the yolk (ECL) for three days starting from 2 dpf whose sensory neurons have been analyzed with limited success due to the lack of adequate molecular and genetic tools to label the neurons in ECL in zebrafish embryo[40]. As shown in Fig. 6a and Supplementary Fig. 18, the gradual increase in fluorescent signals in anastomosing networks in ECL across developmental procedures became evident in embryos with 13xQUAS:ZsGreen at 5 dpf in comparison to that of 5xQUAS:ZsGreen. Given the mesh-like appearance of the fluorescent signals in the yolk (Supplementary Fig. 18) and the equivalent background noise between 5xQUAS:ZsGreen and 13xQUAS:ZsGreen clutches, the observed fluorescence in ECL could not be simply due to the advent of auto-fluorescence or different imaging settings (Fig. 6b). In addition, to minimize the possible emergence of auto-fluorescence in the yolk, we used a fixed confocal setting that did not elicit any perceivable auto-fluorescence when observing the embryos. Taken together, IQ-Switch should be able to tune transgene expression in a spatiotemporal manner via combining different drivers and effectors. In addition, the EQ-On would make enforced reporter gene expression possible even with a relatively weak promoter.

## Discussion

Although there are miscellaneous binary transgene induction cassettes available, one or more inherent weaknesses have hampered their application for the generation of transgenic animals. For instance, Tet-On/Off and stress-inducible promoters are not recommended for conditional gene expression due to the relatively high leakiness of the transgene[4,41]. The GAL4/UAS system has been most successfully applied to zebrafish in transgenesis as well as in enhancer trapping[42,43] owing to its many advantages including a tolerable degree of driver toxicity, low leakiness, reversibility, and a relatively high expression level of transgenes. Furthermore, the versatile heterologous Gal4 drivers conjugated with distinct regulatory modules responsive to their compatible stimuli such as drugs, and the vast depository of transgenic lines under the control of UAS that can be combined by genetic crosses with the discrete Gal4 driver provide a great resource for generating new transgenic zebrafish[3,44,45]. However, cumulative methylation of the UAS element after successive generations results in mosaicism of transgene expression as early as the F3

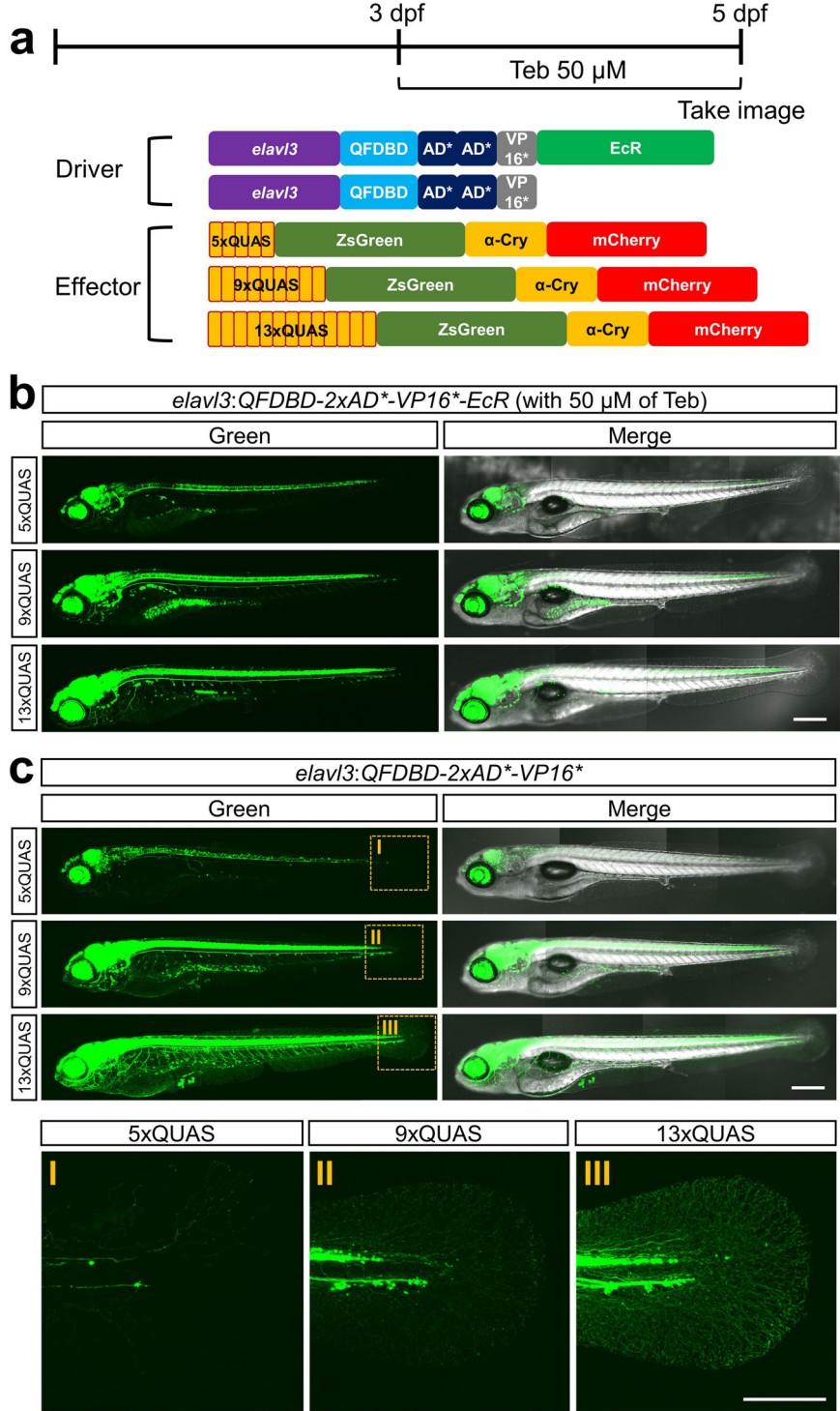

**Fig. 5 Tissue-specific activation of IQ-Switch can be attained by using a discrete cell-type-specific promoter. a** Schematic illustration of a driver under the control of *elavl3* promoter and respective effectors harboring different numbers of QUAS. The driver constructs were equipped with 8.7 kb of *elavl3* promoter instead of *ubb* promoter. Teb (50 μM) was administered for 48 h before imaging at 5 dpf. **b** Genetic cross of F1 *Tg(elavl3:QFDBD-2xAD*-VP16*-EcR)* with indicated F2 effector lines. The siblings were exposed to Teb to induce ZsGreen transgene expression. Note that the intensity of ZsGreen reporter fluorescence in the pan-neuronal tissues driven by *elavl3* promoter was gradually augmented in the order of the increased number of QUAS repeats. **c** Genetic cross of F1 *Tg(elavl3:QFDBD-2xAD*-VP16*)* with indicated F2 effector lines. The intensity of ZsGreen fluorescence among the siblings was most prominent with an effector harboring a 13xQUAS transgene. Abbreviations: Teb; tebufenozide. Scale bar: 200 μm.

generation (Fig. 2e) that may eventually attain complete transcriptional silencing[5].

The methylation-driven transcriptional silencing of transgenes could be overcome by adopting an ITES without containing CpG

dinucleotides in TA-responsive elements as TrpR/tUAS[8] or using a GAL4/UAS system with a limited copy of UAS tandem repeats[5,46]. Although TrpR/tUAS has the virtue of being free from transgene silencing[8], it is regarded as barely suitable for

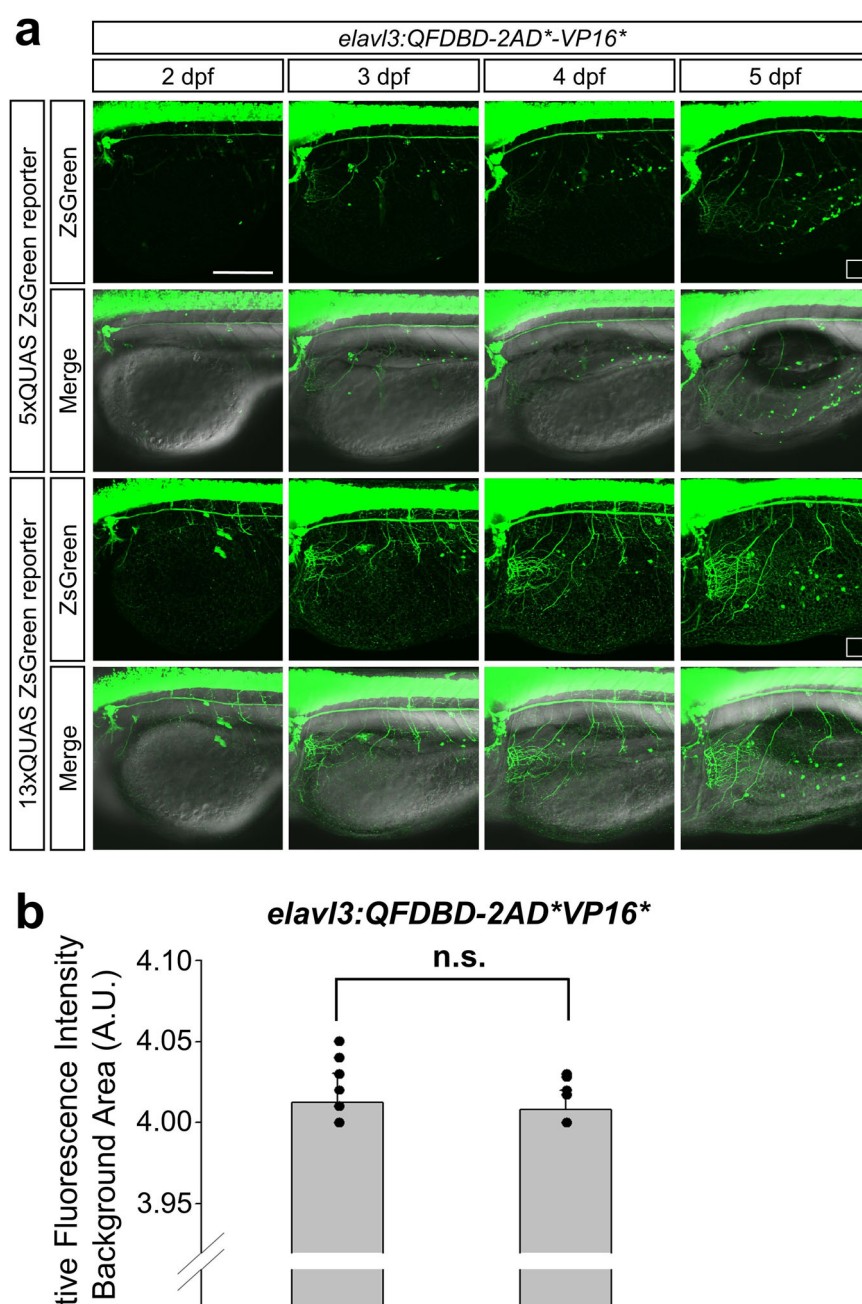

**Fig. 6 The intensity of ZsGreen was gradually mounting on the sensory neurons in EYL across embryonic growth. a** Representative image of embryos at the indicated developmental stage. They obtained from genetic crosses between F2 *Tg(elavl3:QFDBD-2xAD\*-VP16\*)* and F3 effector lines having 5x or 13xQUAS-regulated *ZsGreen* reporter gene. **b** The points labeled with white open square box in (**a**) stand for the region used to level off background fluorescence intensity of 5x and 13xQUAS-ZsGreen groups. Comprehensive level of background fluorescence was measured with 12 single images derived from same number of embryos at 5 dpf under the identical confocal setting. No discernable differences were observed in two discrete clutches. Twelve embryos in each clutch were analyzed in the assay. Abbreviations: A.U.; arbitrary unit, n.s.; not significant, Teb; tebufenozide. Scale bar: 200 µm.

transgenesis without substantial alleviation of the toxic features of TrpR drivers. As previously reported, four copies of the non-repeating UAS construct (4Xnr UAS) were less susceptible to methylation, whereby GAL4 binding properties seem to be retained after three generations[5].

The refractory feature of the Q system against transgene silencing across successive generations[12,13] has drawn much attention from researchers developing new molecular and genetic tools to tackle technical challenges such as methylation-driven promoter inactivation. However, the original Q system[12] could not be adopted as an alternative for generating transgenic zebrafish because of the substantial driver toxicities as shown in Supplementary Fig. 1. Although another QF driver (QF-Gal4) was recently developed as an additional gene switch[14], the potential toxicity of the driver, at least under our experimental conditions, may limit its usage for zebrafish transgenesis

(Supplementary Fig. 2); a reliable QF/QUAS-based gene-tunable cassette suited to vertebrate systems without toxicity has not yet been developed. Therefore, a novel transgene switch with no gene silencing problem and minimal toxicity while retaining all the virtues of the QF/QUAS binary ITES is highly desirable.

To satisfy the aforementioned unmet needs of ITESs, we developed a new binary transgene cassette designated as IQ-Switch, which meets all of the prerequisite criteria for its use in vertebrate models: first, it shows no discernable toxicity or transgene leakiness; second, it shows a several-fold higher responsiveness to the non-toxic chemical inducer Teb compared to the GAL4/UAS system; third, it does not become silenced after its transmission to the next generation, irrespective of its promoter methylation; fourth, the expression level of the transgene can be easily manipulated by altering concentrations of the chemical inducer, or by exploiting effector lines having different numbers of QUAS element tandem repeats; fifth, by using tissue specific promoters, researchers are able to induce the gene of interest in a spatial and temporal manner; sixth, the system is reversible by withdrawal of the chemical inducer; seventh, using a driver that does not have an EcR domain (EQ-On), one can drive transgene expression constitutively by a simple genetic cross with an appropriate effector line without treatment with chemical inducers; eighth, individual effector constructs with different numbers of QUAS repeats could be combined with other potential drivers, for instance QF-Gal4[14], to modulate the yield of transgene expression depending on experimental purposes; and ninth, as a genetic cross with 13xQUAS effector highly potentiated the driver's promoter activity (Fig. 5c), our newly developed gene switch could be applied to transgenesis for the identification of a given promoter's activity in tissues where it is too weak to demarcate itself.

One potential concern of the IQ-Switch is the possible adverse effects of ecdysone analogs on cellular physiology[47–49]. High concentrations of Teb (over 142 μM) induced cell cycle arrest and apoptosis in the human cervical carcinoma cell line HeLa[48,49]; other ecdysone agonists including muristerone A, ponasterone A, and GET™-E inhibited FasL-and TRAIL-induced cell death in the human colon carcinoma cell line RKO[47]. We used Teb at no more than 50 μM in zebrafish embryos; this concentration appears permissive to early embryonic development; however, we could not completely rule out the possibility that Teb concentration might be deleterious to the transgenic animals when they are raised with Teb for a prolonged period. Therefore, it would be desirable to substitute Teb with other ecdysone analogs with less or no cellular toxicity. In addition, for expanding IQ-Switch in broader biological applications, generating a singular plasmid by combining all the requisite components of the driver and effector of IQ-Switch would be helpful, as was the case with a Gal4/UAS-based singular gene switch[4].

The relatively low transactivating activity of IQ-Switch may compare to QF-Gal4 (Supplementary Fig. 16); it appears that although the transactivating activity of QF-Gal4-EcR is similar or higher with Teb treatment than IQ-Switch, and that of QF-Gal4 is higher than that of EQ-On (Supplementary Fig. 16c), QF-Gal4 is more toxic, especially when it is strongly expressed (Supplementary Fig. 2). Thus, IQ-Switch might be a reliable ITES that can balance a relatively high transactivation activity and restrained toxicity. Therefore, IQ-Switch is a beneficial transgene-inducible cassette that can overcome the intrinsic obstacles frequently observed in currently available ITESs. In combination with other gene switches including Gal4/UAS[5,46] and LexPR-LexOP[9,10], the IQ-Switch can serve as an additional and complementary genetic tool to currently available ITESs for the pre-clinical studies of GOF diseases and for elucidating the cellular and embryonic function of genes of interest. In addition, our data strongly suggest that IQ-Switch as well as EQ-On can be applied to other model organisms for the precise control of transgene expression.

## Methods

**Preparation of plasmids for transgenesis**. A Tol2 vector (pKY64-miniTol2-opt-CRY-eFGP) was used as a prime backbone of the plasmid construction (Supplementary Table 2). Individual elements, promoters and all the required elements for generating transgene expression cassette were subcloned into the Tol2 vector (pKY64-miniTol2-opt-CRY-eFGP) using T4 DNA polymerase (NEB)-mediated ligation-independent cloning method following the experimental procedures in a previous report[50]. The *elavl3* promoter from pTol2-elavl3-GCaMP6s (Addgene plasmid #59531) was a gift from Dr. Misha Ahrens. All of the vectors we generated are shown in Supplementary Table 2.

**Zebrafish husbandry and Tol2 mediated Transgenesis**. Zebrafish (*Danio rerio*, AB line) were raised at 28.5 °C with proper water circulation and maintained under a light schedule of 14 h light and 10 h dark. For the generation of transgenic zebrafish, 50 pg of transposase mRNA[51] together with respective plasmids (20 pg each) listed in Supplementary Table 2 were pressure injected at 1-2 cell stages. The in vitro transcribed mRNA encoding transposon[52] was synthesized using mMESSAGE mMACHINE™ SP6 transcription kit (Thermo Fisher Scientific). Among the injected embryos, fluorescent positive siblings in lens at 4 dpf were collected for the husbandry. After one to one outcross of the founder listed in Supplementary Table 3 with a wild type AB line, the siblings positive of EGFP or mCherry fluorescence in lens were collected and raised to adulthood. Three independent lines of each transgenic animal have been maintaining through consecutive outcross with wild type AB line. Zebrafish care was performed in accordance with guidelines from the Korea Research Institute of Bioscience and Biotechnology (KRIBB) and Chungnam National University (201903-CNU-007). All of the transgenic animals we generated are shown in Supplementary Table 3.

**Genomic DNA isolation and bisulfite sequencing**. Tg(*10xUAS:ZsGreen-P2A-braf*$^{V610E}$, *α-cry:EGFP*) and Tg(*5xQUAS:ZsGreen-P2A-BRAF*$^{V600E}$, *α-cry:mCherry*) thirty embryos each at 7dpf were randomly collected and homogenized uniformly. The genomic DNA was isolated using TIANamp Genomic DNA Kit (TIANGEN) following the manufacturer's instruction. Bisulfite sequencing was performed targeting *10xUAS* or *5xQUAS* sequences through company BIONICS.Inc (Korea).

**Total RNA extraction and cDNA synthesis**. For the preparation of total RNA, 30 embryos of each sample were randomly collected at the indicated time points (Fig. 1e). 20 μl of Tri-reagent (Ambion) per embryo was added before homogenized using a 3 ml disposable syringe, and then 1/5 volume of chloroform was added to each sample. The stirred samples were centrifuged (13,000 RPM, 10 min at 4 °C), and then the supernatant was transferred into fresh tube. Total RNAs were precipitated with ice-cold isopropyl alcohol. For the cDNA synthesis, 2.5 μg of total RNA was used for the in vitro reaction with SuperScript reverse transcriptase II (Invitrogen) in accordance with the manufacturer's instructions.

**In vitro transcription**. For the synthesis of mRNA, pCS2+ and pCS2+MT vectors were exploited for the delivery of proper DNA. The vectors were linearized by the digestion with Not I, and then cleared out using a phenol/chloroform extraction. The 1 μg of linearized template were subjected to in vitro reaction using mMES-SAGE mMACHINE™ SP6 Transcription Kit (Invitrogen) following the manufacturer's instruction. The in vitro transcribed mRNA was dissolved in 0.1 M DEPC treated KCl before injection.

**Chromatin immunoprecipitation (ChIP)**. In vitro synthesized mRNAs encoding indicated drivers with N-terminal 6xMyc tag were pressure injected into the progeny from Tg(*5xQUAS:ZsGreen-P2A-BRAF*$^{V600E}$, *α-cry:mCherry*). Embryos at 20-somite stage were fixed with 2.2% paraformaldehyde in egg water (60 μg of sea salt/ml) for 30 min, and then quenched under the 0.125 M glycine. The prepared embryos were homogenized using a disposable syringe in IP buffer (150 mM NaCl, 10 mM Tris-Cl pH 7.5, 1 mM DTT and 0.2% NP40). The embryo lysates were centrifuged (13,000 RPM, 1 min) to obtain nuclear pellets which were subjected to sonication for the preparation of sheared genomic DNA. While 10% of sheared DNA was secured as input control, the rest of the sample was used for ChIP through the incubation with 1 μg of monoclonal anti-Flag (M2, Sigma-Aldrich) or 1 μg of mixtures of anti-Myc antibodies (Santa Cruz Biotechnology, sc-40 plus Millipore, 06-549) for 3 h at 4 °C. The DNA fragments bound by Myc-tagged proteins were obtained using Chelex 100 resin (Bio-Rad) following a protocol[53,54] with a small modification. The amount of precipitated DNA was analyzed using qPCR.

**Quantitative PCR (qPCR)**. For the quantification of the yield of transgene and immunoprecipitated DNA, a TOPreal™ qPCR 2x PreMIX (Enzynomics, Korea) was used for the measurement of amplified samples by exploiting a CFX connect Real-

Time PCR Detection System (Bio-Rad) with Bio-Rad CFX Manager™ 3.1 software for the calibration. The yield of *ZsGreen* was measured in comparison to that of *β-actin*. The assays in Figs. 1e and 2c were carried out in triplicates and error bars stands for the standard error. The calculated data were revalidated using Microsoft Excel. The statistical significance depicted as *p*-value was represented in Fig. 2c. Only the *p*-value less than 0.05 considered as statistically significant.

The following primer sets were used for the detection of the yield of individual samples:

A ZsGreen primer set, for the amplification of 88 bp length of DNA fragment:
*ZsGreen* forward primer, 5′-CATCTTGAAGGGCGACGTGAG-3′;
*ZsGreen* reverse primer, 5′-CTTGGCCTTGTACACGGTGTC-3′;
A *β-actin* primer set, for the amplification of 120 bp length of DNA fragment:
*β-actin* forward primer, 5′-TACAATGAGCTCCGTGTTGCC-3′;
*β-actin* reverse primer, 5′-AGGGGTGTTGAAGGTCTCGAA-3′;
A primer set for targeting QUAS element to amplify 231 bp of DNA fragment:
QUAS amplicon forward primer, 5′-CGAGGTCGACAACTTTGTATAGAAA AGTTG-3′;
QUAS amplicon reverse primer, 5′-TACAAACTTGACGCGTCTTCGAGG-3′.

**Tebufenozide and Vemurafenib treatment**. Tebufenozide (Sigma-Aldrich, 31652) was dissolved in dimethyl sulfoxide (DMSO) at 50 mM stock solution, which is further diluted before treatment to cells or zebrafish embryos in proper culture media. To remove tebufenozide, embryos were washed out several times with fresh egg water (60 μg of sea salt/ml). The vemurafenib (PLX4032; S1267) was purchased from Selleckchem.com (U.S.). Adequate amount of vemurafenib dissolved in DMSO was directly added to the embryos.

**Luciferase assay**. The 5x, 9x, 13x, 17xQUAS and 10xUAS were subcloned into pGL3-Basic vector (Promega). HEK293 cells that were 50% confluent were transfected with 40 ng of pRL-CMV (Promega) together with each 200 ng of driver and effector plasmids using GENE-Fect™ transfection reagent (TransLab Inc, Korea). Luciferase assays were carried out using Dual-Luciferase Reporter Assay System (Promega) following the manufacture's instruction. Each sample was measured at least three times, and significance was assessed by the Student's *t* test. Briefly, the transfectants were cultured for 5 h before treatment of Teb with indicated amount, and then harvested for dual-luciferase assay after further extended culture for another 24 h The statistical significance was represented in Fig. 1f with *p*-value. For *p*-value >0.05, we consider the difference not significant. The standard deviation and Student's *t* test *p*-value were obtained through exploiting Microsoft Excel.

**Quantification of Fluorescence imaging of zebrafish transgenic animals**. To quantify the relative fluorescence intensity of confocal images with different lengths of QUAS tandem repeats, confocal images of the progeny were obtained from genetic crosses of *Tg(ubb:QFDBD-2xAD*-VP16*-EcR)* with transgenic lines integrated with 5x, 9x, and 13x QUAS *ZsGreen* reporter in their genome at 2 dpf using the FV 1000 confocal microscope (Olympus). Before confocal imaging, transgenic zebrafish embryos were incubated with Teb (10 μM) at 1 dpf for 24 h. Zebrafish embryos from the cross of *Tg(ubb:QFDBD-2xAD*-VP16*-EcR)* and Tg with a 13xQUAS *ZsGreen* reporter without Teb incubation were used as negative controls. Quantitative analyses were performed by measuring the relative fluorescence intensity using Image J with the ROI manager (https://imagej.nih.gov/ij/) as previously described[55]. Briefly, Z-projected confocal images were generated by stacking 10 sections with 10 μm thickness. To precisely measure the fluorescence yield, regions of Z-projected images were first divided into three parts (background, brain, and muscle areas), and then the relative fluorescence intensities of the background areas from different samples obtained by Image J with the ROI manager were averaged and set as a basal level (A. U. = 1). Finally, the brain and muscle areas were chosen and the fluorescence intensities of ZsGreen with different tandem repeats of QUAS, obtained by Image J with the ROI manager, relative to the average basal level, were calculated. Notably, the initial relative fluorescence intensities of the background areas of different groups were very similar to each other, confirming that the images were taken under identical conditions (Fig. 4c). Although this comparative analysis was sufficient to reveal the differences among different samples and areas, the fluorescence of the brain area of the 13xQUAS *Zsgreen* reporter line, which was the strongest, tended to be saturated using the current imaging parameters. To circumvent this fluorescence saturation issue, we also measured the fluorescence intensities at single planes of confocal Z-stack images (Supplementary Fig. 12). We chose a single optical section that exhibited the strongest fluorescence intensity out of the total of 10 optical slices of each Z-stack that usually fell into one of the 4th–6th slices. In the case of the negative control without Teb, the 5th single plane was selected and its fluorescence intensities in different areas were measured in the same way as described above. Twenty transgenic zebrafish at 2 dpf per group were measured for quantitative analysis. To show the development of axon tracks in the yolk (Fig. 6 and Supplementary Fig. 18), *Tg(elavl3:QFDBD-2AD*-VP16*)* zebrafish embryos with 5x or 13xQUAS *ZsGreen* reporter were used for imaging. Representative images of embryos at developmental stages from 2 dpf to 5 dpf were obtained daily under the same experimental conditions. To show that images from experiments were taken under identical confocal imaging conditions, the relative fluorescence intensities of the background area of zebrafish embryos with 5x or 13x QUAS *ZsGreen* reporter lines were obtained using Image J with ROI manager as described above and compared. Statistical analyses of the data were performed using the Student's *t* test. The statistical significance is represented in Figs. 4d, 6b and Supplementary Fig. 12, with *p*-values. A *p*-value higher than 0.05 was considered not significant. The standard deviation and Student's *t* test *p*-values were obtained using Microsoft Excel. Fluorescence Z-stack images in Fig. 3 were also obtained by using a CEL-ENA®S digital imaging system (Logos biosystems, Korea).

**Fluorescence imaging of cells**. After transfection of various combination of driver and effector plasmids with appropriate Teb stimuli (10 μM, for 12 h) if necessary, HEK293 and Cos7 transfectants were washed three times with PBS and then fixed with 4% paraformaldehyde for 15 min. Nuclei were stained with 4′, 6-diamidino-2-phenylindole (DAPI; Sigma-Aldrich, D9542) for 2 min. After mounting, fluorescence images were acquired using a confocal laser-scanning microscope (TCS SP8; Leica, Wetzlar, Germany), with constant excitation, emission, pinhole, and exposure time. Equipment parameters were summarized in Supplementary Table 4.

**Western blotting**. After 30 dechorionated zebrafish embryos were rinsed with ice-cold PBS for three times, they were lysed by trituration with a 3 ml disposable syringe in 10 μl of CompLysis™ Protein Extraction Reagent (SignaGen Laboratory, SL100319-L) per embryo with ProEX™ 1x protease and phosphatase inhibitor cocktail (TransLab Inc, Korea, TLP-120I). The lysates were incubated for 15 min on ice, they were subjected to centrifugation for 10 min at $16,000 \times g$ at 4 °C, and then the supernatant was mixed with 5x SDS gel loading buffer before boiling for 5 min. The 12.5 μl of each sample with SDS gel loading buffer being equivalent to the amount of protein extracted from a single embryo was separated in 10% SDS-PAGE, and then transferred into nitrocellulose membrane, 0.45 μm (Pall Corporation). The membrane was immersed in 1x blocking solution (ProNA™ 5x phospho-Block Solution, TransLab Inc, Korea, TLP-115.1 P) at room temperature (RT) for 30 min, incubated with primary antibodies diluted in 1x blocking solution for 8 h at RT. After the membrane was rinsed three times with TBST (10 mM Tris-HCl, 150 mM NaCl, 0.05% Tween-20, pH 8.0), HRP-conjugated secondary antibody diluted in 1x blocking solution was added and then incubated for 1 h at RT. After vigorous washing the membrane with TBST over 5 times, western detection was performed using an ECL detection system supplied from TransLab (ProNA™ ECL Ottimo, TLP-112.1) under the chemiluminescence CCD imaging system (ATTO, Ez-Capture MG). Following antibodies were used in this assay; anti-ZsGreen (Takara Bio Clontech, 632474), anti-actin (Sigma-Aldrich, A2066), goat anti-rabbit IgG secondary antibody-HRP (Thermo Fisher Scientific, 31460).

**Statistics and reproducibility**. All the luciferase assays and qPCR measurements were carried out triplicate or more. In the case of western blotting, three independent experiments had been analyzed using Image J before calculating *p*-value. Only the calculated *p*-value less than 0.05 considered as statistically valid. ZsGreen fluorescence intensity in zebrafish embryos using Image J was calibrated in comparison with their own background intensity. The number of embryos analyzed was depicted in appropriate figure legends.

**Reporting summary**. Further information on research design is available in the Nature Research Reporting Summary linked to this article.

## Data availability

The raw data for graphs is available in supplementary data 1. Any further information can be obtained from corresponding authors upon reasonable request. The sequence information of the core vectors used in this study is shown in supplementary data 2, and the detailed information of them will be available in GenBank and Addgene.

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

## Acknowledgements

The authors really appreciate Dr. Igor B. Dawid (NIH-NICHD, US) for the critical reading and valuable comments on the manuscript. This work was supported by the National Research Foundation of Korea (NRF) grant funded by the Korea government (NRF-2016R1D1A3B01010007, No.2020R1A2C2005317, NRF-2019R1A2C1087934, and NRF-2019R1A2C1010661), by National Research Council of Science & Technology (NST) of Ministry of Science and ICT of Korea (CRC-15-04-KIST), and by KRIBB Research Initiative Program (KGM5352113, KGM2112133).

## Author contributions

J.H., J.-G.L., and K.-C.S. equally contributed to the paper by carrying out construction of the plasmids, generation of transgenic animals, ChIP, qPCR, western blotting, luciferase assays, data processing and confocal imaging of zebrafish embryos. K.L. also contributed heavily to the construction of the plasmids and the generation of transgenic animals. H.S.J., and D.-K.C. took confocal images of animal cells. J.L., J.H., D.C., and Y.H. were involved in generation of transgenic zebrafish carrying a mutated *lamin A (Δ37)*. S.L., S.U.L., Y.K., J.J., Y.-K.B., and R.H.H. contributed to the construction of plasmids and the maintenance of transgenic animals. G.M.H., J.-S.L., and H.R. are co-corresponding authors who had supervised the whole process of the experiments and had written the manuscript.

## Competing interests

The authors declare no competing interests.

## Additional information

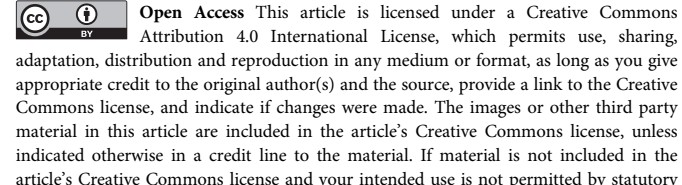

