## [Transparent Peer Review File · Communications Biology]

Reviewers' comments:

Reviewer #1 (Remarks to the Author):

In this work, Hong et al designed a new gene switch system, called IQ-Switch, that has many advantages over the commonly-used strategies, such as Gal4-UAS, Tet on/off with low toxicity/leakage, and no transgene silencing. They also used this approach to develop a disease model, which might facilitate future drug screening and/or mechanistic studies.

Overall, the work is very well designed and the results are quite convincing and reliable, and also the manuscript is very well written. I only have very minor concerns on some of this work. There are a lack of enough statistics data and the significance value should be provided in most of experiments with graphs (Fig 1, 4, S8). Even regarding the fluorescence intensity, for example Fig S7, it would be better to provide some quantitative data to show the comparison etc.

Reviewer #2 (Remarks to the Author):

The work by Hong et al. describes a new transgene expression system, IQ-Switch, and its variants and characterize their applications in cell culture and zebrafish systems. The authors show that IQ-Switch system has several advantages over the previous inducible transgene expression switches, including the low toxicity, tunability of expression levels and low levels of gene silencing. The described genetic tool has a broad potential for applications and therefore of interest to the zebrafish research community. The writing is generally clear and accessible even to non-experts.

Despite the strength mentioned above, the paper has several weaknesses. The authors took the impressive effort to test many kinds of experimental conditions, namely different driver variants, QUAS reporters, as well as induction parameters for the tebufenozide (Teb) application. In other words, however, the paper contains a lot of information and some important controls and appropriate comparisons are missing at places, which together makes it hard to interpret their results. Particularly, the application of Teb is performed for different durations in various experiments presented in this paper without any explanation as to why these particular durations were chosen. It is not only confusing but also makes it difficult to compare between different experiments and appreciate the real potential of the IQ-Switch. Overall, it is essential for the authors to address these issues so that readers can correctly assess the validity of the present study.

Major points

1. The authors claim that the IQ-Switch is a tunable gene expression system. The dose dependent increase of the expression level was shown in cultured cells (Fig. 1d), but not for zebrafish embryos (only inexplicitly in Fig. 4d and supplementary Fig. 6). Please provide a side-by-side comparison of the expression levels of QUAS reporter driven by IQ-Switch driver induced with a various concentration of Teb, including a 0 μ M Teb control. The 0 μ M Teb control is essential because it addresses the leakiness of the system.
2. To induce the expression of IQ-Switch in zebrafish embryos, Teb was applied for different durations in each experiment throughout the study. The durations should be kept consistent across different experiments as much as possible. If there are reasons why the authors decided to use the particular duration, please explain these reasons either in the main text, methods or figure legend.
3. Throughout the paper, the authors rely on the fluorescent intensity imaged by confocal microscopy to quantify the expression levels in zebrafish embryos. The fluorescent intensity from the confocal images are generally not suited for quantification, since the signals are highly dependent on the settings of the confocal imaging, such as laser intensity, detector gain, pinhole etc. In particular, the saturated images (such as the one shown in Fig. 4d and 5b) does not allow proper quantification.

For example, the authors concluded that ZsGreen intensity became saturated for 9xQUAS and thus did not find a difference with 16xQUAS in (Fig. 4d). It is not clear whether (a) the ZsGreen expression reached a plateau in the 9xQUAS case and it does not increase further in the 16xQUAS case or (b) there is actually a difference in the expression levels between the two reporters but the detection of the fluorescent signals is already saturated with 9xQUAS and cannot detect more intense signal for the 16xQUAS. Another example is in Fig. 5c, where the authors conclude that the ZsGreen reporter expression is highest in the 13xQUAS line. If all three images were taken with the same confocal settings, why are there brighter signals in the yolk region of the larva in 13xQUAS? These signals are certainly not derived from the neuron-specific *elavl3* promoter and therefore should be equally detected in all three conditions if they were imaged in a same way.

To address these issues, I would request the following.

First, in the method describing the confocal imaging of cultured cells in "Fluorescence Imaging" section, the authors write that the same imaging settings were used for different groups of the same experiments. Please clarify if the same principle was applied for the imaging of zebrafish embryos (in the method, they describe that the laser intensity was maintained constant, but no mention about other parameters).

Furthermore, the authors should try at least another approach to better quantify the expression levels in zebrafish embryos. Ideally, flow cytometry should be used to measure the fluorescence intensity, such as done in Burgess et al., *Dev. Biol.*, 2020. Alternatively, instead of a stack of a thick volume of tissue, a thinner subvolume (or even a single optical plane) should be presented to avoid saturated images and better represent the expression levels.

4. The paper emphasizes the reversibility of IQ-Switch. To evaluate the reversibility of IQ-Switch, the authors measured the mRNA expression of the ZsGreen reporter gene by RT-qPCR after the washout of Teb. The off-kinetics of the reporter mRNA expression seems quite slow (the reporter mRNA level is reduced to only about 70% of the original expression 24hr after the Teb removal). Please discuss a potential cause for this slow kinetics. In addition, the authors do not show the downregulation of the expression at the protein level. It can be imagined that the ZsGreen protein is quite stable and therefore the protein downregulation (or fluorescence decrease) is difficult to detect. If so, another kind of protein should be used.

5. Another claim that the authors make is the low degree of silencing. The data show that a) QUAS is less methylated than UAS, b) IQ-Switch can bind to the 5xQUAS (although please also see my comment about the ChIP above) and c) IQ-Switch induces more uniform and higher transgene expression than its Gal4/UAS equivalent. Although the data indicate that their new IQ-Switch system is less silenced than Gal4 system, it is not clear if this low level of silencing is specifically attributed to the IQ-Switch or it is a general feature for all QF-based systems. To address this, the ChIP (Fig. 2b,c) and the transgene expression analysis in embryos (Fig. 2d) should be done using other QF variants (for example QF, QF2, QF-Gal4).

6. They found that a variant of IQ-Switch that lacks the EcR (namely QFDBD-2xAD*-VP16*) acts as a constitutively active form and show that it can be used as a potent driver. However, it is not entirely clear to me whether this form is superior to other QF drivers that have been previously reported, such as QF, QF2 and QF-Gal4 i.e. how novel is this QFDBD-2xAD*-VP16* variant? As far as I can tell, the toxicity of QFDBD-2xAD*-VP16* variant is compared with the previously reported QF-Gal4AD (Burgess et al., 2020) (Supplementary Fig. 1), but the transactivation level of the reporter is not directly compared with the previous QF variants.

7. In the ChIP analysis, samples were collected 8 hours after Teb was removed. It is not clear why they performed the ChIP assay after Teb is already washed out, when the IQ-Switch protein cannot supposedly stay in the nucleus in the absence of Teb.

8. The previous study by Esengil et al., *Nat Chemical Biology* 2007 has used the Gal4VP16-EcR system and is similar to the design principle of the present work. However, I feel that this work is not

described well enough. In order for the reader to assess the novelty of the current study, the authors should discuss this work in the Introduction, before they describe their own design of IQ-switch.

9. The first half of the Discussion describes the past research without discussing any of the new results obtained from this study. This part should be moved to Introduction. Instead, please discuss potential limitations of the system.

Minor points

1. On page 5, the authors state that "In mammalian cells and zebrafish embryos the driver did not elicit significant toxicity, irrespective of Teb treatments (Supplementary Fig. 1 and Supplementary Table 1). ", but there is no toxicity data on mammalian cells.

2. For the schematic shown in Fig. 2d, a promoter is missing for the Gal4DBD-VP16*-EcR.

3. The word (Q)UAS "enhancer" was frequently used in this manuscript. The term "enhancer" generally refers to a sequence element that enhanced the expression of a gene encoded in the genome, and is not usually used for the (Q)UAS.

4. In Fig. 2d, the authors conclude that the IQ-Switch/QUAS system can induce more uniform expression of the transgene (ZsGreen and BRAF protein) than the Gal4/UAS counterpart can. The embryo obtained from the cross of Gal4/10xUAS (D2 + E2) shows ZsGreen-positive cell aggregates in the ventral side of the embryos. Please explain what they are. Was it caused by the expression of the BRAF gene?

5. Again in Fig. 2d, the QUAS reporter and UAS reporter express different types of the BRAF gene ("BRAF V600E" for QUAS and "Braf V610E" for UAS). Since two different proteins are expressed in each case, the authors cannot directly compare the degree of malformations observed in the two conditions and speculate the expression levels based on the different degrees on the malformation, unless the function of these two BRAF genes are identical. They should use the same BRAF protein subtype for both reporter lines to assess the degree of malformations.

6. The constitutive active form of IQ-Switch (a variant without EcR) is also called "switch" (page 5, 2nd line from bottom). The term "switch" is generally used for drug inducible systems or switch of transgenes upon a recombination event by recombinases such as Cre. Therefore, in my opinion, the term "switch" is not appropriate for the constitutive active driver.

Reviewer #3 (Remarks to the Author):

In this manuscript, the author developed a novel inducible gene expression system by combining the EcR(ecdysone receptor)-Teb system with the QF/QUAS system in zebrafish. First, the authors constructed more than 10 possible driver constructs and selected a suitable one from the aspects of toxicity and inducibility using HEK and zebrafish. Then they indeed demonstrated the induction of gene expression in transgenic zebrafish. Second, the authors applied the technique to overexpression of an activated form of BRAF and showed successful demonstration of a drug effect. Third, the authors optimized an effector construct by changing the length of QUAS. The manuscript is concise and clearly written.

Major points:

(1) The author disgraded the Gal4/UAS system too much, that has been widely used in zebrafish and lead to successful hundreds of publications. The authors better focus on what are new in the manuscript.

(1)-1: Citations about the Gal4-UAS system in zebrafish are poor and inappropriate. Authors' evaluation of the Gal4-UAS system in zebrafish is pretty much biased.

(1)-2: There was a report describing the silencing. But for other cases silencing with the Gal4-UAS system has not been that problematic.

(1)-3: In most successful cases, 5xUAS has been used (for instance, see *Advances in Genetics*, 95: 65-87 (2016)). Therefore, comparison of QUAS with 10xUAS does not make sense.

(2) As for the experiments using transgenic fish, quantitative and statistical analyses are lacking.

(2)-1: The authors showed only images of zebrafish. The photos may be champion results. The quantitative analyses such as qPCR, determination of the numbers of transposon insertions transgenic fish, etc, are needed.

(2)-2: Statistical analysis (how many fish are analyzed?) should be needed.

(3) The authors, in many places in the text, overstated their achievements. The QF/QUAS system in zebrafish has been already described. What is new here is a combination between the QF/QUAS system and EcR-Teb system. The authors need to write the manuscript for readers to understand this. The word "innovative" in the title sounds a little odd to me.

(3)-1: It is not clear whether QUAS will never be silenced after tens of generation.

(3)-2: Apparently, QF/QUAS is an addition to zebrafish methodology, but not alternative to Gal4-UAS.

(3)-3: The precise explanation and evaluation of the previous works related to QF/QUAS is poor.

Minor point:

In the method section, I do not see how they made transgenic fish (Tol2 trans genesis).

Rebuttal letter

Reviewer 1.

In this work, Hong et al designed a new gene switch system, called IQ-Switch, that has many advantages over the commonly-used strategies, such as Gal4-UAS, Tet on/off with low toxicity/leakage, and no transgene silencing. They also used this approach to develop a disease model, which might facilitate future drug screening and/or mechanistic studies.

1. Overall, the work is very well designed and the results are quite convincing and reliable, and also the manuscript is very well written. I only have very minor concerns on some of this work. There are a lack of enough statistics data and the significance value should be provided in most of experiments with graphs (Fig 1, 4, S8).

→ We apologize for the lack of statistical analysis of some data. As the Reviewer pointed it out, we have performed appropriate statistical analyses and indicated the statistical relevance in all graphs.

2. Even regarding the fluorescence intensity, for example Fig S7, it would be better to provide some quantitative data to show the comparison etc.

→ Thanks for the advice. For the better comparison of the induction level of ZsGreen reporter under the control of discrete 5x, 9x, and 13xQUAS regulatory elements with or without administration of tebufenozide, we quantified the fluorescent intensity of ZsGreen using image J in the brain and muscle in respective transgenic animals with a fixed confocal setting. The representative figure is shown in Figure 4c, 4d and Supplementary Figure 12. In addition, we also carried out western blot experiments to quantitatively analyze the responsiveness of QUAS to the different dosage of Teb (Supplementary Fig. 4) as well as the increased transgene expression in the order of QUAS copies by the stimuli of a fixed concentration of Teb (Supplementary Fig. 14).

Reviewer #2 (Remarks to the Author):

The work by Hong et al. describes a new transgene expression system, IQ-Switch, and its variants and characterize their applications in cell culture and zebrafish systems. The authors show that IQ-Switch system has several advantages over the previous inducible transgene expression switches, including the low toxicity, tunability of expression levels and low levels of gene silencing. The described genetic tool has a broad potential for applications and therefore of interest to the zebrafish research community. The writing is generally clear and accessible even to non-experts.

Despite the strength mentioned above, the paper has several weaknesses. The authors took the

impressive effort to test many kinds of experimental conditions, namely different driver variants, QUAS reporters, as well as induction parameters for the tebufenozide (Teb) application. In other words, however, the paper contains a lot of information and some important controls and appropriate comparisons are missing at places, which together makes it hard to interpret their results. Particularly, the application of Teb is performed for different durations in various experiments presented in this paper without any explanation as to why these particular durations were chosen. It is not only confusing but also makes it difficult to compare between different experiments and appreciate the real potential of the IQ-Switch. Overall, it is essential for the authors to address these issues so that readers can correctly assess the validity of the present study.

Major points

1. The authors claim that the IQ-Switch is a tunable gene expression system. The dose dependent increase of the expression level was shown in cultured cells (Fig. 1d), but not for zebrafish embryos (only inexplicitly in Fig. 4d and supplementary Fig. 6). Please provide a side-by-side comparison of the expression levels of QUAS reporter driven by IQ-Switch driver induced with a various concentration of Teb, including a 0 μM Teb control. The 0 μM Teb control is essential because it addresses the leakiness of the system.

→ Thanks for the comments. We added data in order to show the tebufenozide (Teb) concentration dependent inducibility of IQ-Switch in zebrafish embryos. As shown in supplementary figure 4, we validated by western blotting the Teb dosage dependent increment of the ZsGreen protein level using total embryonic lysates. Importantly, we did not detect any observable leakiness of the IQ-Switch without treatment of Teb.

2. To induce the expression of IQ-Switch in zebrafish embryos, Teb was applied for different durations in each experiment throughout the study. The durations should be kept consistent across different experiments as much as possible. If there are reasons why the authors decided to use the particular duration, please explain these reasons either in the main text, methods or figure legend

→ Thanks for the valuable comment. We described reasonable explanations as to why we chose different time points of Teb treatment and the duration of incubation in the main text and figure legends. In brief, when we induced toxic transgene for instance BRAF(V600E), we chose two different strategies in order to minimize gross embryonic lethality. A high concentration of Teb (50 μM) was administered during early embryonic developmental stage with only two-hour duration, while a low dose of Teb (2.5 μM) was treated for prolonged time period at from 36 hpf in order to mimic RASopathy disease with BRAF(V600E). To prevent confusion which could ensue as the reviewer indicated, we have carefully described this issue in the main text in the revised manuscript (see the 2nd module of the Results entitled 'Methylatio -independent activation of IQ-Switch).

3. Throughout the paper, the authors rely on the fluorescent intensity imaged by confocal microscopy to quantify the expression levels in zebrafish embryos. The fluorescent intensity from the confocal images are generally not suited for quantification, since the signals are highly dependent on the settings of the confocal imaging, such as laser intensity, detector gain, pinhole etc. In particular, the saturated images (such as the one shown in Fig. 4d and 5b) does not allow proper quantification. For example, the authors concluded that ZsGreen intensity became saturated for 9xQUAS and thus did not find a difference with 16xQUAS in (Fig. 4d). It is not clear whether (a) the ZsGreen expression reached a plateau in the 9xQUAS case and it does not increase further in the 16xQUAS case or (b) there is actually a difference in the expression levels between the two reporters but the detection of the fluorescent signals is already saturated with 9xQUAS and cannot detect more intense signal for the 16xQUAS. Another example is in Fig. 5c, where the authors conclude that the ZsGreen reporter expression is highest in the 13xQUAS line. If all three images were taken with the same confocal settings, why are there brighter signals in the yolk region of the larva in 13xQUAS? These signals are certainly not derived from the neuron-specific *elavl3* promoter and therefore should be equally detected in all three conditions if they were imaged in a same way.

→ To quantify the inducibility of IQ-Switch following different concentrations of tebufenozide as well as the increased number of QUAS regulatory elements, we carried out western blotting using whole embryonic lysates. The data were added in supplementary figure 4 and supplementary figure 14. In addition, we analyzed the fluorescent intensities with a fixed settings of confocal microscope in the brain and muscle of multiple embryos ($n = 20$) and then quantified the images using Image J software (Fig.4c,d and Supplementary Fig. 12). The detailed experimental methods and imaging parameters used are described in Methods section. Collectively, the data clearly showed that the sensitivity of IQ-Switch was dependent upon the dosage of tebufenozide and the increased number of QUAS reached the maximum level at 13xQUAS.

In the case of the fluorescence in the yolk, to prevent the possible auto-fluorescence in yolk from arising, we chose a fixed confocal setting (see Methods, Fluorescence Imaging) which did not elicit any detectable auto-fluorescent in yolk at 2 dpf and then repeatedly imaged the same embryos with identical setting until they reached at 5 dpf. Since the fluorescence in yolk became gradually detectable only in 13xQUAS transgenic animals as the embryos developed under the same imaging conditions (Fig. 6), we concluded that the appearance of fluorescence in yolk was very unlikely simply due to the background in 13xQUAS. Rather, the fluorescence in the yolk appears to represent the complicated arborization of differentiating neurons in the epithelial cell layers in the yolk (Supplementary Fig. 18), which has not been reported in *elavl* promoter (8.7 kb)-driven transgenic animals but became observable with 13xQUAS due to the amplification of the fluorescent signal. In comparison to a reference as below, the gradually accentuated fluorescent signal in the yolk following embryonic

development under the control of *e/avl* promoter-driven EQ-On driver and 13xQUAS-ZsGreen effector represents growing sensory axons whereupon they finally form anastomosing neuronal networks. We cited the reference in the main text.

“At embryonic and early larval stages, the zebrafish skin consists of two epithelial cell layers, the outer periderm and inner basal cell layer; the basal cell layer is separated from underlying tissues by a laminin-rich basement membrane” (O’Brien et al., 2011 JCN “Coordinate Development of Skin Cells and Cutaneous Sensory Axons in zebrafish”).

To address these issues, I would request the following.

First, in the method describing the confocal imaging of cultured cells in “Fluorescence Imaging” section, the authors write that the same imaging settings were used for different groups of the same experiments. Please clarify if the same principle was applied for the imaging of zebrafish embryos (in the method, they describe that the laser intensity was maintained constant, but no mention about other parameters).

→ We described all the parameters of the confocal setting in the methods section (see Methods, “Quantification of Fluorescence imaging of zebrafish transgenic animals”, and Supplementary Table 4 which described the confocal setting for mammalian cell imaging).

Furthermore, the authors should try at least another approach to better quantify the expression levels in zebrafish embryos. Ideally, flow cytometry should be used to measure the fluorescence intensity, such as done in Burgess et al., Dev. Biol., 2020. Alternatively, instead of a stack of a thick volume of tissue, a thinner subvolume (or even a single optical plane) should be presented to avoid saturated images and better represent the expression levels.

→ Thanks for the valuable comments. For the precise quantification and comparison of the transgene expression, we measured that level by western blotting (Supplementary Fig. 4 and Supplementary Fig. 14) as well as confocal single optical plane images (Supplementary Figure 12) as well as Z-stack (Fig. 4) using image J and compared the differences of the fluorescent intensities in 5xQUAS, 9xQUAS, and 13xQUAS transgenic animals. The detailed parameters and methodology for the quantification and comparison were described in a section “Fluorescence Imaging and Quantification of Fluorescence imaging of zebrafish transgenic animals” in Methods.

4. The paper emphasizes the reversibility of IQ-Switch. To evaluate the reversibility of IQ-Switch, the authors measured the mRNA expression of the ZsGreen reporter gene by RT-qPCR after the washout of Teb. The off-kinetics of the reporter mRNA expression seems quite slow (the reporter mRNA level is reduced to only about 70% of the original expression 24hr after the Teb removal). Please discuss a potential cause for this slow kinetics. In addition, the authors do not show the downregulation of the expression at the protein level. It can be imagined that the ZsGreen protein is quite stable and therefore the protein downregulation (or fluorescence decrease) is difficult to detect. If so, another kind of protein should be used.

→ We thank the Reviewer for raising this important point. The relative slow off-kinetics is a general feature of steroid-related chemical inducers such as tebufenozide, since they need to be completely metabolized in cells or tissues for clearing out. The results are consistent with our previous report (Lee, S. et al. Ecdysone Receptor-based Singular Gene Switches for Regulated Transgene Expression in Cells and Adult Rodent Tissues. *Mol Ther Nucleic Acids* **5**, e367 (2016)). In addition, as Reviewer #2 pointed out, ZsGreen protein was gradually accumulating in embryos even after complete washing of tebufenozide presumably due to the high stability of ZsGreen. Thus, we measured the mRNA level of ZsGreen instead of its final translated product that may precisely reflect the reversible nature of IQ-Switch with even slower pharmacokinetics. Here we presented a western blot figure as below (only for Reviewer), and addressed explanation in the main text as to the slow kinetics with citing appropriate references. Rational explanation of this issue was discussed in Result section (1st module of Results entitled ‘Refining a novel QF driver that is sensitive to exogenous chemical stimuli’)

(Figure only for Reviewer). Highly stable ZsGreen protein had constantly accumulated in embryos even after washing out the Teb. The larvae were obtained from genetic crosses between F3 driver *Tg(ubb:QFDBD-2xAD*-VP16*-EcR)* and F3 effector *Tg(QUAS:ZsGreen-P2A)*. At 24 hpf, 50 μM of Teb was added in the embryonic culture medium for 12 hours. After cleaning out the Teb, the embryos had been moved to fresh petri-dish and cultured in fresh medium for the indicated time period.

5. Another claim that the authors make is the low degree of silencing. The data show that a) QUAS is less methylated than UAS, b) IQ-Switch can bind to the 5xQUAS (although please also see my comment about the ChIP above) and c) IQ-Switch induces more uniform and higher transgene expression than its Gal4/UAS equivalent. Although the data indicate that their new IQ-Switch system is less silenced than Gal4 system, it is not clear if this low level of silencing is specifically attributed to the IQ-Switch or it is a general feature for all QF-based systems. To address this, the ChIP (Fig. 2b,c) and the transgene expression analysis in embryos (Fig. 2d) should be done using other QF variants (for example QF, QF2, QF-Gal4).

→ Following the suggestion from the Reviewer, we had carried out new ChIP experiment with QF-Gal4 driver (QFDBD-Gal4AD-EcR), where an identical enrichment of the QF-Gal4 driver on 5xQUAS was confirmed as that of the IQ-Switch driver (QFDBD-2xAD*-VP16*-EcR, Supplementary Fig. 6), suggesting that methylation-independent binding of QFDBD to QUAS elements is a general feature for the QF-based system. Unfortunately, however, we were unable to perform the same experiments with other driver constructs, such as original QF driver and QF-ΔMD (QFDBD-AD-EcR), due to their unacceptable toxicity on early embryonic development (Supplementary Fig. 1).

6. They found that a variant of IQ-Switch that lacks the EcR (namely QFDBD-2xAD*-VP16*) acts as a constitutively active form and show that it can be used as a potent driver. However, it is not entirely clear to me whether this form is superior to other QF drivers that have been previously reported, such as QF,

QF2 and QF-Gal4 i.e. how novel is this QFDBD-2xAD*-VP16* variant? As far as I can tell, the toxicity of QFDBD-2xAD*-VP16* variant is compared with the previously reported QF-Gal4AD (Burgess et al., 2020) (Supplementary Fig. 1), but the transactivation level of the reporter is not directly compared with the previous QF variants.

→ We agree with the Reviewer that it would be more informative and consolidating if we provide evidence of the transactivation potential of our IQ-Switch by comparing with other QF drivers that have been previously reported. To address this issue, we directly compared the transactivation level of QFDBD-2xAD*-VP16* with that of previously reported QF-Gal4AD in Supplementary Figure 16c. In brief, while Teb dependent activity of luciferase reporter did not show any differences between QFDBD-2xAD*-VP16*-EcR and QF-Gal4AD-EcR at above 13xQUAS, constitutive active form of QF-Gal4AD was around 1.5-fold stronger than that of QFDBD-2xAD*-VP16*. However, as we mentioned in the main text, when we considered the cellular toxicity, the QFDBD-2xAD*-VP16* seemed to put less cellular burden than did QF-Gal4AD-EcR (Supplementary Fig. 2).

7. In the ChIP analysis, samples were collected 8 hours after Teb was removed. It is not clear why they performed the ChIP assay after Teb is already washed out, when the IQ-Switch protein cannot supposedly stay in the nucleus in the absence of Teb.

→ Thanks for pointing out the disputable problem. As shown in Figure 1e, tebufenozide seemed to lingering for a relatively long period in treated cells after washing, the transgene expression was maintained for a fairly long time, steadily declining. In addition, a significant toxicity of BRAF(V600E) interfered with early embryonic development with an extended treatment of tebufenozide at high doses. Therefore, to circumvent the early lethality problem we treated with a high concentration of tebufenozide for a short period and then raised them until they showed expected embryonic developmental defects with limited lethality and ZsGreen expression, of which phenotypes became obvious 8 hours after Teb removal. The raised issues were discussed in Results section (1st and 2nd module of Results)

8. The previous study by Esengil et al., Nat Chemical Biology 2007 has used the Gal4VP16-EcR system and is similar to the design principle of the present work. However, I feel that this work is not described well enough. In order for the reader to assess the novelty of the current study, the authors should discuss this work in the Introduction, before they describe their own design of IQ-switch.

→ We thank the Reviewer for this valuable comment. We cited and explained the paper by Esengil et al., (2017) in the Introduction. For the better readability of our manuscript, we added more comprehensive data (one more main figure and nine more supplementary figures) and explanation to emphasize the novelty of IQ-Switch in the Results and Discussion.

9. The first half of the Discussion describes the past research without discussing any of the new results obtained from this study. This part should be moved to Introduction. Instead, please discuss potential limitations of the system.

→ We thank the Reviewer for bring this issue to our attention. Accordingly, we have revised the Introduction and Discussion sections of manuscript. We also described potential limitations of IQ-Switch in Discussion part, with citing appropriate references.

Minor points

1. On page 5, the authors state that “In mammalian cells and zebrafish embryos the driver did not elicit significant toxicity, irrespective of Teb treatments (Supplementary Fig. 1 and Supplementary Table 1). “, but there is no toxicity data on mammalian cells.

→ Thanks for the pointing out the problem of our ambiguous description. As we did not validate the toxicity of IQ-Switch in the mammalian cells, we have changed the sentence to “In zebrafish embryos the driver did not elicit significant toxicity, irrespective of Teb treatments (Supplementary Fig. 1 and Supplementary Table 1)” on page 5.

2. For the schematic shown in Fig. 2d, a promoter is missing for the Gal4DBD-VP16*-EcR.

→ We appreciated the detail comment. We amended the Figure 2d.

3. The word (Q)UAS “enhancer” was frequently used in this manuscript. The term “enhancer” generally refers to a sequence element that enhanced the expression of a gene encoded in the genome, and is not usually used for the (Q)UAS.

→ In response to reviewer, we changed the term “enhancer” to “element”.

4. In Fig. 2d, the authors conclude that the IQ-Switch/QUAS system can induce more uniform expression of the transgene (ZsGreen and BRAF protein) than the Gal4/UAS counterpart can. The embryo obtained from the cross of Gal4/10xUAS (D2 + E2) shows ZsGreen-positive cell aggregates in the ventral side of the embryos. Please explain what they are. Was it caused by the expression of the BRAF gene?

→ In response to reviewer’s inquiry, we carried out genetic crosses again with transgenic lines of more generations passed down, and observed still uniform expression of ZsGreen-P2A-BRAF(V600E) in embryos obtained from a genetic cross of F3 driver with F6 effector of IQ-Switch (Fig. 2d,e and

Supplementary Fig. 7). In contrast, the transgene silencing effect under the control of 10xUAS was evident to us from embryos obtained from a genetic cross of Gal4DBD-VP16*-EcR (F2) and 10xUAS-ZsGreen-P2A-Braf(V610E) (F2) lines. ZsGreen expression in this F2 cross with tebufenozide stimuli initially appeared as aggregates as shown in figure 2d,e and supplementary figure 7. In F3 generation of the 10xUAS-ZsGreen-P2A-Braf(V610E) effector embryos, the ZsGreen reporter gene was no longer detected as aggregation in the yolk, but instead showed a salt-pepper appearance in a few cells indicating a significant/severe gene silencing (Fig. 2e and Supplementary Fig. 7b). We substituted the new data for older ones in figure 2 and described the new result in main text and figure legends.

5. Again in Fig. 2d, the QUAS reporter and UAS reporter express different types of the BRAF gene ("BRAF V600E" for QUAS and "Braf V610E" for UAS). Since two different proteins are expressed in each case, the authors cannot directly compare the degree of malformations observed in the two conditions and speculate the expression levels based on the different degrees on the malformation, unless the function of these two BRAF genes are identical. They should use the same BRAF protein subtype for both reporter lines to assess the degree of malformations.

→ We agreed that we should have generated BRAF(V600E) TG under the control of 10xUAS for the direct comparison between IQ-Switch and Gal4-UAS system. Though we now have been raising F0 embryos harboring zebrafish Braf(V610E) IQ-Switch effector line, it is too early stage to compare both transgene switches under the identical condition. However, since both effector lines share a similar structural features composed of ZsGreen-P2A-transgene (Fig. 2d and Supplementary Figure 7a), we believe that our experimental design would be good enough for the purpose of analyzing transgene silencing over the generations. In any case we would like to apologize to reviewers for rendering inadvertent confusion.

6. The constitutive active form of IQ-Switch (a variant without EcR) is also called "switch" (page 5, 2nd line from bottom). The term "switch" is generally used for drug inducible systems or switch of transgenes upon a recombination event by recombinases such as Cre. Therefore, in my opinion, the term "switch" is not appropriate for the constitutive active driver.

→ Thanks for the comment. Instead the term "constitutive active form of IQ-Switch", we decided to use EQ-On (Everlasting QF transgene switch-On).

Reviewer #3 (Remarks to the Author):

In this manuscript, the author developed a novel inducible gene expression system by combining the EcR(ecdysone receptor)-Teb system with the QF/QUAS system in zebrafish. First, the authors constructed more than 10 possible driver constructs and selected a suitable one from the aspects of

toxicity and inducibility using HEK and zebrafish. Then they indeed demonstrated the induction of gene expression in transgenic zebrafish. Second, the authors applied the technique to overexpression of an activated form of BRAF and showed successful demonstration of a drug effect. Third, the authors optimized an effector construct by changing the length of QUAS. The manuscript is concise and clearly written.

Major points:

(1) The author disgraced the Gal4/UAS system too much, that has been widely used in zebrafish and lead to successful hundreds of publications. The authors better focus on what are new in the manuscript.

(1)-1: Citations about the Gal4-UAS system in zebrafish are poor and inappropriate. Authors' evaluation of the Gal4-UAS system in zebrafish is pretty much biased.

(1)-2: There was a report describing the silencing. But for other cases silencing with the Gal4-UAS system has not been that problematic.

(1)-3: In most successful cases, 5xUAS has been used (for instance, see *Advances in Genetics*, 95: 65-87 (2016)). Therefore, comparison of QUAS with 10xUAS does not make sense.

→ We thank the Reviewer for bring this issue to our attention. As the Reviewer kindly pointed it out, we rewrote our manuscript since the Gal4-UAS system has been widely and successfully used in the field of zebrafish society with minimal limitations. We did not intend to belittle the GAL4-UAS system, of which the contribution has made an enormous progress in zebrafish researches. Thus, our IQ-Switch would be an additional and compensatory tool for the generation of transgenic animals. Following reviewer's advices, we added proper references including "*Advances in Genetics*, 95: 65-87 (2016)" in our main text. We appreciated again the valuable comments.

(2) As for the experiments using transgenic fish, quantitative and statistical analyses are lacking.

(2)-1: The authors showed only images of zebrafish. The photos may be champion results. The quantitative analyses such as qPCR, determination of the numbers of transposon insertions transgenic fish, etc, are needed.

→ In response to reviewer, for more precise quantification of our results, we carried out several rounds of western blotting (Supplementary Fig. 4 and Supplementary Fig. 14) using whole embryonic lysate to detect the translational level of ZsGreen reporter in comparison to that of α -actin as a loading control, and Image J quantification with fixed values of confocal setting (Fig. 4c,e, Fig. 6 and Supplementary Fig. 18, see the Methods, "Fluorescence Imaging and Quantification of Fluorescence imaging of zebrafish transgenic animals"). qPCR was also performed to determine the reversibility of IQ-Switch with appropriate statistical analyses with p-values (Fig. 1e). Other qPCRs set out to test the enrichment of QFDBD on 5xQUAS elements with p-value annotations (Fig. 2c and Supplementary Fig. 6).

(2)-2: Statistical analysis (how many fish are analyzed?) should be needed.

→ Thanks for pointing out the omission of the statistical analysis. We added proper statistical data in each figure wherever necessary including the number of fish and embryos analyzed (Fig. 3w, Fig. 4d, Fig. 6b, Supplementary Fig. 1 and Supplementary Fig. 2, see figure legends). In addition, we clarified the number of F0 adulthood of each transgenic animals in supplementary table 2.

(3) The authors, in many places in the text, overstated their achievements. The QF/QUAS system in zebrafish has been already described. What is new here is a combination between the QF/QUAS system and EcR-Teb system. The authors need to write the manuscript for readers to understand this. The word "innovative" in the title sounds a little odd to me.

→ Thanks for the comment. In response to reviewer, we changed the title as to "IQ-Switch: QF-based innocuous, silencing-free, and inducible gene switch system in zebrafish". We also rewrote the Introduction and Discussion to clarify that the new achievement here is based upon the previously reported performances to hurdle some technical challenges not completely solved yet. In addition, to our best knowledge the QFDBD-2xAD*-VP16*-EcR configuration together with further extended and optimized QUAS repeats has never been tested and validated in animal transgenesis.

(3)-1: It is not clear whether QUAS will never be silenced after tens of generation.

→ In response to reviewer, we acknowledge that in the current study, we could not monitor the possible occurrence of transgene silencing of IQ-Switch after over ten generations within this specified revision period due to the time limitation. However, we did not see any gene silencing at least at seventh generation of the transgenic embryos obtained from a genetic cross of the F3 driver (Ubb-QFDBD-2xAD*-VP16*-EcR) with the F6 effector (5xQUAS-ZsGreen-P2A-BRAF(V600E) (Fig. 2e and Supplementary Fig. 7b). We will maintain and carefully scrutinize the transgenic fish lines in order to see whether the transgene may be silenced at any points over generations.

(3)-2: Apparently, QF/QUAS is an addition to zebrafish methodology, but not alternative to Gal4-UAS.

→ Since we agreed the reviewer's point, we rewrote the Discussion section.

(3)-3: The precise explanation and evaluation of the previous works related to QF/QUAS is poor.

→ In response to reviewer, we described more detail explanation about the previous works and related

references of the QF-QUAS in the Introduction and Discussion section.

Minor point:

In the method section, I do not see how they made transgenic fish (Tol2 trans genesis).

→ Thanks for the point. We generated the transgenic fish using Tol2 transgenesis. We added the detail experimental procedure with proper references in Methods section.

REVIEWERS' COMMENTS:

Reviewer #2 (Remarks to the Author):

The authors have properly addressed my concerns and the revised manuscript can be accepted for publication.

Reviewer #3 (Remarks to the Author):

I found that the authors well responded to my comments and revised the manuscript. I would like to request the following points.

In abstract:

"is free of driver toxicity and transgene silencing"

this should be weakened since the author did not know this system is perfectly free of these. The authors may say "relatively" or add "at least up to seventh generations"

line 81:

"have been hampering"

is too strong since many expression systems have been constructed based on these.

I suggest "have been potentially problematic" etc.

line 108:

"non-silencing feature"

is too strong. I suggest "potentially non-silencing feature"

line 113:

"free of gene silencing"

is too strong. I suggest "gene silencing at least up to seventh generations" etc.

line 363:

"eventually attains"> "may eventually attain"

line 367:

Here the authors already mentioned "a limited copy of.."

Therefore the next part is unnecessary.

Remove from line 372 "Instead..." to line 380 "reliable gal4/UAS system"

Rebuttal letter

Reviewer 1.

In this work, Hong et al designed a new gene switch system, called IQ-Switch, that has many advantages over the commonly-used strategies, such as Gal4-UAS, Tet on/off with low toxicity/leakage, and no transgene silencing. They also used this approach to develop a disease model, which might facilitate future drug screening and/or mechanistic studies.

1. Overall, the work is very well designed and the results are quite convincing and reliable, and also the manuscript is very well written. I only have very minor concerns on some of this work. There are a lack of enough statistics data and the significance value should be provided in most of experiments with graphs (Fig 1, 4, S8).

→ We apologize for the lack of statistical analysis of some data. As the Reviewer pointed it out, we have performed appropriate statistical analyses and indicated the statistical relevance in all graphs.

2. Even regarding the fluorescence intensity, for example Fig S7, it would be better to provide some quantitative data to show the comparison etc.

→ Thanks for the advice. For the better comparison of the induction level of ZsGreen reporter under the control of discrete 5x, 9x, and 13xQUAS regulatory elements with or without administration of tebufenozide, we quantified the fluorescent intensity of ZsGreen using image J in the brain and muscle in respective transgenic animals with a fixed confocal setting. The representative figure is shown in Figure 4c, 4d and Supplementary Figure 12. In addition, we also carried out western blot experiments to quantitatively analyze the responsiveness of QUAS to the different dosage of Teb (Supplementary Fig. 4) as well as the increased transgene expression in the order of QUAS copies by the stimuli of a fixed concentration of Teb (Supplementary Fig. 14).

Reviewer #2 (Remarks to the Author):

The work by Hong et al. describes a new transgene expression system, IQ-Switch, and its variants and characterize their applications in cell culture and zebrafish systems. The authors show that IQ-Switch system has several advantages over the previous inducible transgene expression switches, including the low toxicity, tunability of expression levels and low levels of gene silencing. The described genetic tool has a broad potential for applications and therefore of interest to the zebrafish research community. The writing is generally clear and accessible even to non-experts.

Despite the strength mentioned above, the paper has several weaknesses. The authors took the

impressive effort to test many kinds of experimental conditions, namely different driver variants, QUAS reporters, as well as induction parameters for the tebufenozide (Teb) application. In other words, however, the paper contains a lot of information and some important controls and appropriate comparisons are missing at places, which together makes it hard to interpret their results. Particularly, the application of Teb is performed for different durations in various experiments presented in this paper without any explanation as to why these particular durations were chosen. It is not only confusing but also makes it difficult to compare between different experiments and appreciate the real potential of the IQ-Switch. Overall, it is essential for the authors to address these issues so that readers can correctly assess the validity of the present study.

Major points

1. The authors claim that the IQ-Switch is a tunable gene expression system. The dose dependent increase of the expression level was shown in cultured cells (Fig. 1d), but not for zebrafish embryos (only inexplicitly in Fig. 4d and supplementary Fig. 6). Please provide a side-by-side comparison of the expression levels of QUAS reporter driven by IQ-Switch driver induced with a various concentration of Teb, including a 0 μ M Teb control. The 0 μ M Teb control is essential because it addresses the leakiness of the system.

→ Thanks for the comments. We added data in order to show the tebufenozide (Teb) concentration dependent inducibility of IQ-Switch in zebrafish embryos. As shown in supplementary figure 4, we validated by western blotting the Teb dosage dependent increment of the ZsGreen protein level using total embryonic lysates. Importantly, we did not detect any observable leakiness of the IQ-Switch without treatment of Teb.

2. To induce the expression of IQ-Switch in zebrafish embryos, Teb was applied for different durations in each experiment throughout the study. The durations should be kept consistent across different experiments as much as possible. If there are reasons why the authors decided to use the particular duration, please explain these reasons either in the main text, methods or figure legend

→ Thanks for the valuable comment. We described reasonable explanations as to why we chose different time points of Teb treatment and the duration of incubation in the main text and figure legends. In brief, when we induced toxic transgene for instance BRAF(V600E), we chose two different strategies in order to minimize gross embryonic lethality. A high concentration of Teb (50 μ M) was administered during early embryonic developmental stage with only two-hour duration, while a low dose of Teb (2.5 μ M) was treated for prolonged time period at from 36 hpf in order to mimic RASopathy disease with BRAF(V600E). To prevent confusion which could ensue as the reviewer indicated, we have carefully described this issue in the main text in the revised manuscript (see the 2nd module of the Results entitled 'Methylatio -independent activation of IQ-Switch).

3. Throughout the paper, the authors rely on the fluorescent intensity imaged by confocal microscopy to quantify the expression levels in zebrafish embryos. The fluorescent intensity from the confocal images are generally not suited for quantification, since the signals are highly dependent on the settings of the confocal imaging, such as laser intensity, detector gain, pinhole etc. In particular, the saturated images (such as the one shown in Fig. 4d and 5b) does not allow proper quantification. For example, the authors concluded that ZsGreen intensity became saturated for 9xQUAS and thus did not find a difference with 16xQUAS in (Fig. 4d). It is not clear whether (a) the ZsGreen expression reached a plateau in the 9xQUAS case and it does not increase further in the 16xQUAS case or (b) there is actually a difference in the expression levels between the two reporters but the detection of the fluorescent signals is already saturated with 9xQUAS and cannot detect more intense signal for the 16xQUAS. Another example is in Fig. 5c, where the authors conclude that the ZsGreen reporter expression is highest in the 13xQUAS line. If all three images were taken with the same confocal settings, why are there brighter signals in the yolk region of the larva in 13xQUAS? These signals are certainly not derived from the neuron-specific *elavl3* promoter and therefore should be equally detected in all three conditions if they were imaged in a same way.

→ To quantify the inducibility of IQ-Switch following different concentrations of tebufenozide as well as the increased number of QUAS regulatory elements, we carried out western blotting using whole embryonic lysates. The data were added in supplementary figure 4 and supplementary figure 14. In addition, we analyzed the fluorescent intensities with a fixed settings of confocal microscope in the brain and muscle of multiple embryos (n = 20) and then quantified the images using Image J software (Fig.4c,d and Supplementary Fig. 12). The detailed experimental methods and imaging parameters used are described in Methods section. Collectively, the data clearly showed that the sensitivity of IQ-Switch was dependent upon the dosage of tebufenozide and the increased number of QUAS reached the maximum level at 13xQUAS.

In the case of the fluorescence in the yolk, to prevent the possible auto-fluorescence in yolk from arising, we chose a fixed confocal setting (see Methods, Fluorescence Imaging) which did not elicit any detectable auto-fluorescent in yolk at 2 dpf and then repeatedly imaged the same embryos with identical setting until they reached at 5 dpf. Since the fluorescence in yolk became gradually detectable only in 13xQUAS transgenic animals as the embryos developed under the same imaging conditions (Fig. 6), we concluded that the appearance of fluorescence in yolk was very unlikely simply due to the background in 13xQUAS. Rather, the fluorescence in the yolk appears to represent the complicated arborization of differentiating neurons in the epithelial cell layers in the yolk (Supplementary Fig. 18), which has not been reported in *elavl* promoter (8.7 kb)-driven transgenic animals but became observable with 13xQUAS due to the amplification of the fluorescent signal. In comparison to a reference as below, the gradually accentuated fluorescent signal in the yolk following embryonic

development under the control of *e/avl* promoter-driven EQ-On driver and 13xQUAS-ZsGreen effector represents growing sensory axons whereupon they finally form anastomosing neuronal networks. We cited the reference in the main text.

“At embryonic and early larval stages, the zebrafish skin consists of two epithelial cell layers, the outer periderm and inner basal cell layer; the basal cell layer is separated from underlying tissues by a laminin-rich basement membrane” (O’Brien et al., 2011 JCN “Coordinate Development of Skin Cells and Cutaneous Sensory Axons in zebrafish”).

To address these issues, I would request the following.

First, in the method describing the confocal imaging of cultured cells in “Fluorescence Imaging” section, the authors write that the same imaging settings were used for different groups of the same experiments. Please clarify if the same principle was applied for the imaging of zebrafish embryos (in the method, they describe that the laser intensity was maintained constant, but no mention about other parameters).

→ We described all the parameters of the confocal setting in the methods section (see Methods, “Quantification of Fluorescence imaging of zebrafish transgenic animals”, and Supplementary Table 4 which described the confocal setting for mammalian cell imaging).

Furthermore, the authors should try at least another approach to better quantify the expression levels in zebrafish embryos. Ideally, flow cytometry should be used to measure the fluorescence intensity, such as done in Burgess et al., Dev. Biol., 2020. Alternatively, instead of a stack of a thick volume of tissue, a thinner subvolume (or even a single optical plane) should be presented to avoid saturated images and better represent the expression levels.

→ Thanks for the valuable comments. For the precise quantification and comparison of the transgene expression, we measured that level by western blotting (Supplementary Fig. 4 and Supplementary Fig. 14) as well as confocal single optical plane images (Supplementary Figure 12) as well as Z-stack (Fig. 4) using image J and compared the differences of the fluorescent intensities in 5xQUAS, 9xQUAS, and 13xQUAS transgenic animals. The detailed parameters and methodology for the quantification and comparison were described in a section “Fluorescence Imaging and Quantification of Fluorescence imaging of zebrafish transgenic animals” in Methods.

4. The paper emphasizes the reversibility of IQ-Switch. To evaluate the reversibility of IQ-Switch, the authors measured the mRNA expression of the ZsGreen reporter gene by RT-qPCR after the washout of Teb. The off-kinetics of the reporter mRNA expression seems quite slow (the reporter mRNA level is reduced to only about 70% of the original expression 24hr after the Teb removal). Please discuss a potential cause for this slow kinetics. In addition, the authors do not show the downregulation of the expression at the protein level. It can be imagined that the ZsGreen protein is quite stable and therefore the protein downregulation (or fluorescence decrease) is difficult to detect. If so, another kind of protein should be used.

→ We thank the Reviewer for raising this important point. The relative slow off-kinetics is a general feature of steroid-related chemical inducers such as tebufenozide, since they need to be completely metabolized in cells or tissues for clearing out. The results are consistent with our previous report (Lee, S. et al. Ecdysone Receptor-based Singular Gene Switches for Regulated Transgene Expression in Cells and Adult Rodent Tissues. *Mol Ther Nucleic Acids* **5**, e367 (2016)). In addition, as Reviewer #2 pointed out, ZsGreen protein was gradually accumulating in embryos even after complete washing of tebufenozide presumably due to the high stability of ZsGreen. Thus, we measured the mRNA level of ZsGreen instead of its final translated product that may precisely reflect the reversible nature of IQ-Switch with even slower pharmacokinetics. Here we presented a western blot figure as below (only for Reviewer), and addressed explanation in the main text as to the slow kinetics with citing appropriate references. Rational explanation of this issue was discussed in Result section (1st module of Results entitled ‘Refining a novel QF driver that is sensitive to exogenous chemical stimuli’)

(Figure only for Reviewer). Highly stable ZsGreen protein had constantly accumulated in embryos even after washing out the Teb. The larvae were obtained from genetic crosses between F3 driver *Tg(ubb:QFDBD-2xAD*-VP16*-EcR)* and F3 effector *Tg(QUAS:ZsGreen-P2A)*. At 24 hpf, 50 μM of Teb was added in the embryonic culture medium for 12 hours. After cleaning out the Teb, the embryos had been moved to fresh petri-dish and cultured in fresh medium for the indicated time period.

5. Another claim that the authors make is the low degree of silencing. The data show that a) QUAS is less methylated than UAS, b) IQ-Switch can bind to the 5xQUAS (although please also see my comment about the ChIP above) and c) IQ-Switch induces more uniform and higher transgene expression than its Gal4/UAS equivalent. Although the data indicate that their new IQ-Switch system is less silenced than Gal4 system, it is not clear if this low level of silencing is specifically attributed to the IQ-Switch or it is a general feature for all QF-based systems. To address this, the ChIP (Fig. 2b,c) and the transgene expression analysis in embryos (Fig. 2d) should be done using other QF variants (for example QF, QF2, QF-Gal4).

→ Following the suggestion from the Reviewer, we had carried out new ChIP experiment with QF-Gal4 driver (QFDBD-Gal4AD-EcR), where an identical enrichment of the QF-Gal4 driver on 5xQUAS was confirmed as that of the IQ-Switch driver (QFDBD-2xAD*-VP16*-EcR, Supplementary Fig. 6), suggesting that methylation-independent binding of QFDBD to QUAS elements is a general feature for the QF-based system. Unfortunately, however, we were unable to perform the same experiments with other driver constructs, such as original QF driver and QF-ΔMD (QFDBD-AD-EcR), due to their unacceptable toxicity on early embryonic development (Supplementary Fig. 1).

6. They found that a variant of IQ-Switch that lacks the EcR (namely QFDBD-2xAD*-VP16*) acts as a constitutively active form and show that it can be used as a potent driver. However, it is not entirely clear to me whether this form is superior to other QF drivers that have been previously reported, such as QF,

QF2 and QF-Gal4 i.e. how novel is this QFDBD-2xAD*-VP16* variant? As far as I can tell, the toxicity of QFDBD-2xAD*-VP16* variant is compared with the previously reported QF-Gal4AD (Burgess et al., 2020) (Supplementary Fig. 1), but the transactivation level of the reporter is not directly compared with the previous QF variants.

→ We agree with the Reviewer that it would be more informative and consolidating if we provide evidence of the transactivation potential of our IQ-Switch by comparing with other QF drivers that have been previously reported. To address this issue, we directly compared the transactivation level of QFDBD-2xAD*-VP16* with that of previously reported QF-Gal4AD in Supplementary Figure 16c. In brief, while Teb dependent activity of luciferase reporter did not show any differences between QFDBD-2xAD*-VP16*-EcR and QF-Gal4AD-EcR at above 13xQUAS, constitutive active form of QF-Gal4AD was around 1.5-fold stronger than that of QFDBD-2xAD*-VP16*. However, as we mentioned in the main text, when we considered the cellular toxicity, the QFDBD-2xAD*-VP16* seemed to put less cellular burden than did QF-Gal4AD-EcR (Supplementary Fig. 2).

7. In the ChIP analysis, samples were collected 8 hours after Teb was removed. It is not clear why they performed the ChIP assay after Teb is already washed out, when the IQ-Switch protein cannot supposedly stay in the nucleus in the absence of Teb.

→ Thanks for pointing out the disputable problem. As shown in Figure 1e, tebufenozide seemed to lingering for a relatively long period in treated cells after washing, the transgene expression was maintained for a fairly long time, steadily declining. In addition, a significant toxicity of BRAF(V600E) interfered with early embryonic development with an extended treatment of tebufenozide at high doses. Therefore, to circumvent the early lethality problem we treated with a high concentration of tebufenozide for a short period and then raised them until they showed expected embryonic developmental defects with limited lethality and ZsGreen expression, of which phenotypes became obvious 8 hours after Teb removal. The raised issues were discussed in Results section (1st and 2nd module of Results)

8. The previous study by Esengil et al., Nat Chemical Biology 2007 has used the Gal4VP16-EcR system and is similar to the design principle of the present work. However, I feel that this work is not described well enough. In order for the reader to assess the novelty of the current study, the authors should discuss this work in the Introduction, before they describe their own design of IQ-switch.

→ We thank the Reviewer for this valuable comment. We cited and explained the paper by Esengil et al., (2017) in the Introduction. For the better readability of our manuscript, we added more comprehensive data (one more main figure and nine more supplementary figures) and explanation to emphasize the novelty of IQ-Switch in the Results and Discussion.

9. The first half of the Discussion describes the past research without discussing any of the new results obtained from this study. This part should be moved to Introduction. Instead, please discuss potential limitations of the system.

→ We thank the Reviewer for bring this issue to our attention. Accordingly, we have revised the Introduction and Discussion sections of manuscript. We also described potential limitations of IQ-Switch in Discussion part, with citing appropriate references.

Minor points

1. On page 5, the authors state that “In mammalian cells and zebrafish embryos the driver did not elicit significant toxicity, irrespective of Teb treatments (Supplementary Fig. 1 and Supplementary Table 1). “, but there is no toxicity data on mammalian cells.

→ Thanks for the pointing out the problem of our ambiguous description. As we did not validate the toxicity of IQ-Switch in the mammalian cells, we have changed the sentence to “In zebrafish embryos the driver did not elicit significant toxicity, irrespective of Teb treatments (Supplementary Fig. 1 and Supplementary Table 1)” on page 5.

2. For the schematic shown in Fig. 2d, a promoter is missing for the Gal4DBD-VP16*-EcR.

→ We appreciated the detail comment. We amended the Figure 2d.

3. The word (Q)UAS “enhancer” was frequently used in this manuscript. The term “enhancer” generally refers to a sequence element that enhanced the expression of a gene encoded in the genome, and is not usually used for the (Q)UAS.

→ In response to reviewer, we changed the term “enhancer” to “element”.

4. In Fig. 2d, the authors conclude that the IQ-Switch/QUAS system can induce more uniform expression of the transgene (ZsGreen and BRAF protein) than the Gal4/UAS counterpart can. The embryo obtained from the cross of Gal4/10xUAS (D2 + E2) shows ZsGreen-positive cell aggregates in the ventral side of the embryos. Please explain what they are. Was it caused by the expression of the BRAF gene?

→ In response to reviewer’s inquiry, we carried out genetic crosses again with transgenic lines of more generations passed down, and observed still uniform expression of ZsGreen-P2A-BRAF(V600E) in embryos obtained from a genetic cross of F3 driver with F6 effector of IQ-Switch (Fig. 2d,e and

Supplementary Fig. 7). In contrast, the transgene silencing effect under the control of 10xUAS was evident to us from embryos obtained from a genetic cross of Gal4DBD-VP16*-EcR (F2) and 10xUAS-ZsGreen-P2A-Braf(V610E) (F2) lines. ZsGreen expression in this F2 cross with tebufenozide stimuli initially appeared as aggregates as shown in figure 2d,e and supplementary figure 7. In F3 generation of the 10xUAS-ZsGreen-P2A-Braf(V610E) effector embryos, the ZsGreen reporter gene was no longer detected as aggregation in the yolk, but instead showed a salt-pepper appearance in a few cells indicating a significant/severe gene silencing (Fig. 2e and Supplementary Fig. 7b). We substituted the new data for older ones in figure 2 and described the new result in main text and figure legends.

5. Again in Fig. 2d, the QUAS reporter and UAS reporter express different types of the BRAF gene ("BRAF V600E" for QUAS and "Braf V610E" for UAS). Since two different proteins are expressed in each case, the authors cannot directly compare the degree of malformations observed in the two conditions and speculate the expression levels based on the different degrees on the malformation, unless the function of these two BRAF genes are identical. They should use the same BRAF protein subtype for both reporter lines to assess the degree of malformations.

→ We agreed that we should have generated BRAF(V600E) TG under the control of 10xUAS for the direct comparison between IQ-Switch and Gal4-UAS system. Though we now have been raising F0 embryos harboring zebrafish Braf(V610E) IQ-Switch effector line, it is too early stage to compare both transgene switches under the identical condition. However, since both effector lines share similar structural features composed of ZsGreen-P2A-transgene (Fig. 2d and Supplementary Figure 7a), we believe that our experimental design would be good enough for the purpose of analyzing transgene silencing over the generations. In any case we would like to apologize to reviewers for rendering inadvertent confusion.

6. The constitutive active form of IQ-Switch (a variant without EcR) is also called "switch" (page 5, 2nd line from bottom). The term "switch" is generally used for drug inducible systems or switch of transgenes upon a recombination event by recombinases such as Cre. Therefore, in my opinion, the term "switch" is not appropriate for the constitutive active driver.

→ Thanks for the comment. Instead the term "constitutive active form of IQ-Switch", we decided to use EQ-On (Everlasting QF transgene switch-On).

Reviewer #3 (Remarks to the Author):

In this manuscript, the author developed a novel inducible gene expression system by combining the EcR(ecdysone receptor)-Teb system with the QF/QUAS system in zebrafish. First, the authors constructed more than 10 possible driver constructs and selected a suitable one from the aspects of

toxicity and inducibility using HEK and zebrafish. Then they indeed demonstrated the induction of gene expression in transgenic zebrafish. Second, the authors applied the technique to overexpression of an activated form of BRAF and showed successful demonstration of a drug effect. Third, the authors optimized an effector construct by changing the length of QUAS. The manuscript is concise and clearly written.

Major points:

(1) The author disgraced the Gal4/UAS system too much, that has been widely used in zebrafish and lead to successful hundreds of publications. The authors better focus on what are new in the manuscript.

(1)-1: Citations about the Gal4-UAS system in zebrafish are poor and inappropriate. Authors' evaluation of the Gal4-UAS system in zebrafish is pretty much biased.

(1)-2: There was a report describing the silencing. But for other cases silencing with the Gal4-UAS system has not been that problematic.

(1)-3: In most successful cases, 5xUAS has been used (for instance, see *Advances in Genetics*, 95: 65-87 (2016)). Therefore, comparison of QUAS with 10xUAS does not make sense.

→ We thank the Reviewer for bring this issue to our attention. As the Reviewer kindly pointed it out, we rewrote our manuscript since the Gal4-UAS system has been widely and successfully used in the field of zebrafish society with minimal limitations. We did not intend to belittle the GAL4-UAS system, of which the contribution has made an enormous progress in zebrafish researches. Thus, our IQ-Switch would be an additional and compensatory tool for the generation of transgenic animals. Following reviewer's advices, we added proper references including "*Advances in Genetics*, 95: 65-87 (2016)" in our main text. We appreciated again the valuable comments.

(2) As for the experiments using transgenic fish, quantitative and statistical analyses are lacking.

(2)-1: The authors showed only images of zebrafish. The photos may be champion results. The quantitative analyses such as qPCR, determination of the numbers of transposon insertions transgenic fish, etc, are needed.

→ In response to reviewer, for more precise quantification of our results, we carried out several rounds of western blotting (Supplementary Fig. 4 and Supplementary Fig. 14) using whole embryonic lysate to detect the translational level of ZsGreen reporter in comparison to that of α -actin as a loading control, and Image J quantification with fixed values of confocal setting (Fig. 4c,e, Fig. 6 and Supplementary Fig. 18, see the Methods, "Fluorescence Imaging and Quantification of Fluorescence imaging of zebrafish transgenic animals"). qPCR was also performed to determine the reversibility of IQ-Switch with appropriate statistical analyses with p-values (Fig. 1e). Other qPCRs set out to test the enrichment of QFDBD on 5xQUAS elements with p-value annotations (Fig. 2c and Supplementary Fig. 6).

(2)-2: Statistical analysis (how many fish are analyzed?) should be needed.

→ Thanks for pointing out the omission of the statistical analysis. We added proper statistical data in each figure wherever necessary including the number of fish and embryos analyzed (Fig. 3w, Fig. 4d, Fig. 6b, Supplementary Fig. 1 and Supplementary Fig. 2, see figure legends). In addition, we clarified the number of F0 adulthood of each transgenic animals in supplementary table 2.

(3) The authors, in many places in the text, overstated their achievements. The QF/QUAS system in zebrafish has been already described. What is new here is a combination between the QF/QUAS system and EcR-Teb system. The authors need to write the manuscript for readers to understand this. The word "innovative" in the title sounds a little odd to me.

→ Thanks for the comment. In response to reviewer, we changed the title as to "IQ-Switch: QF-based innocuous, silencing-free, and inducible gene switch system in zebrafish". We also rewrote the Introduction and Discussion to clarify that the new achievement here is based upon the previously reported performances to hurdle some technical challenges not completely solved yet. In addition, to our best knowledge the QFDBD-2xAD*-VP16*-EcR configuration together with further extended and optimized QUAS repeats has never been tested and validated in animal transgenesis.

(3)-1: It is not clear whether QUAS will never be silenced after tens of generation.

→ In response to reviewer, we acknowledge that in the current study, we could not monitor the possible occurrence of transgene silencing of IQ-Switch after over ten generations within this specified revision period due to the time limitation. However, we did not see any gene silencing at least at seventh generation of the transgenic embryos obtained from a genetic cross of the F3 driver (Ubb-QFDBD-2xAD*-VP16*-EcR) with the F6 effector (5xQUAS-ZsGreen-P2A-BRAF(V600E) (Fig. 2e and Supplementary Fig. 7b). We will maintain and carefully scrutinize the transgenic fish lines in order to see whether the transgene may be silenced at any points over generations.

(3)-2: Apparently, QF/QUAS is an addition to zebrafish methodology, but not alternative to Gal4-UAS.

→ Since we agreed the reviewer's point, we rewrote the Discussion section.

(3)-3: The precise explanation and evaluation of the previous works related to QF/QUAS is poor.

→ In response to reviewer, we described more detail explanation about the previous works and related

references of the QF-QUAS in the Introduction and Discussion section.

Minor point:

In the method section, I do not see how they made transgenic fish (Tol2 trans genesis).

→ Thanks for the point. We generated the transgenic fish using Tol2 transgenesis. We added the detail experimental procedure with proper references in Methods section.

REVIEWERS' COMMENTS:

Reviewer #2 (Remarks to the Author):

The authors have properly addressed my concerns and the revised manuscript can be accepted for publication.

Reviewer #3 (Remarks to the Author):

I found that the authors well responded to my comments and revised the manuscript.

I would like to request the following points.

In abstract:

"is free of driver toxicity and transgene silencing"

this should be weakened since the author did not know this system is perfectly free of these. The authors may say "relatively" or add "at least up to seventh generations"

→ Thanks for the advice. We changed our manuscript following reviewer's comments.

line 81:

"have been hampering"

is too strong since many expression systems have been constructed based on these.

I suggest "have been potentially problematic" etc.

→ Thanks for the advice. We changed our manuscript following reviewer's comments.

line 108:

"non-silencing feature"

is too strong. I suggest "potentially non-silencing feature"

→ Thanks for the advice. We changed our manuscript following reviewer's comments.

line 113:

"free of gene silencing"

is too strong. I suggest "gene silencing at least up to seventh generations" etc.

→ Thanks for the advice. We changed our manuscript following reviewer's comments.

line 363:

"eventually attains"> "may eventually attain"

→ Thanks for the advice. We changed our manuscript following reviewer's comments.

line 367:

Here the authors already mentioned "a limited copy of.."

Therefore the next part is unnecessary.

Remove from line 372 "Instead..." to line 380 "reliable gal4/UAS system"

→ Thanks for the advice. We changed our manuscript following reviewer's comments.